# Murine- and Human-Derived Autologous Organoid/Immune Cell Co-Cultures as Pre-Clinical Models of Pancreatic Ductal Adenocarcinoma

**DOI:** 10.3390/cancers12123816

**Published:** 2020-12-17

**Authors:** Loryn Holokai, Jayati Chakrabarti, Joanne Lundy, Daniel Croagh, Pritha Adhikary, Scott S. Richards, Chantal Woodson, Nina Steele, Robert Kuester, Aaron Scott, Mohammad Khreiss, Timothy Frankel, Juanita Merchant, Brendan J. Jenkins, Jiang Wang, Rachna T. Shroff, Syed A. Ahmad, Yana Zavros

**Affiliations:** 1Department of Molecular Genetics, Biochemistry and Microbiology, University of Cincinnati, Cincinnati, OH 45220, USA; holokall@mail.uc.edu (L.H.); woodsocs@mail.uc.edu (C.W.); 2Department of Cellular and Molecular Medicine, University of Arizona, Tucson, AZ 85719, USA; chakraj@email.arizona.edu (J.C.); prithaadhikary@email.arizona.edu (P.A.); 3Centre for Innate Immunity and Infectious Diseases, Hudson Institute of Medical Research, Clayton, VIC 3168, Australia; joanne.lundy@monash.edu (J.L.); brendan.jenkins@hudson.org.au (B.J.J.); 4Department of Molecular Translational Science, School of Clinical Sciences, Monash University, Clayton, VIC 3800, Australia; 5Department of Surgery, School of Clinical Sciences at Monash Health, Monash University, Clayton, VIC 3800, Australia; Daniel.Croagh@monash.edu; 6Department of Gastroenterology and Hepatology, University of Arizona College of Medicine, Tucson, AZ 85719, USA; scottrichards@arizona.edu (S.S.R.); kuester@deptofmed.arizona.edu (R.K.); jmerchant@deptofmed.arizona.edu (J.M.); 7Department of Surgery, University of Michigan, Ann Arbor, MI 48109, USA; steelen@med.umich.edu (N.S.); timofran@med.umich.edu (T.F.); 8Division of Hematology and Oncology, University of Arizona College of Medicine, Tucson, AZ 85719, USA; ajscott@arizona.edu (A.S.); mkhreiss@surgery.arizona.edu (M.K.); rshroff@arizona.edu (R.T.S.); 9Department of Pathology and Laboratory Medicine, University of Cincinnati College of Medicine, Cincinnati, OH 45267, USA; wajn@ucmail.uc.edu; 10Department of Surgery, Division of Surgical Oncology, University of Cincinnati, Cincinnati, OH 45221, USA; ahmadsy@ucmail.uc.edu

**Keywords:** pancreatic ductal adenocarcinoma (PDAC), myeloid derived suppressor cells (MDSCs), PD-L1, organoids, organoid/immune-cell co-culture

## Abstract

**Simple Summary:**

Pancreatic ductal adenocarcinoma (PDAC) is one of the most lethal malignancies, with an approximate 10% five-year survival rate despite therapy. A plausible reason for this observation may be that other, redundant, immune-suppressive mechanisms are at play. Thus, effective treatment of PDAC is a medical challenge and warrants the development of a pre-clinical model whereby the patient’s tumor immune phenotype is characterized and the immune response within the tumor microenvironment tested prior to therapy. These studies present a pre-clinical organoid model that may be used to test the efficacy of combinatorial therapies and targeted therapies, based on modulating the tumor microenvironment, to improve cancer patient response and survival.

**Abstract:**

*Purpose*: Pancreatic ductal adenocarcinoma (PDAC) has the lowest five-year survival rate of all cancers in the United States. Programmed death 1 receptor (PD-1)-programmed death ligand 1 (PD-L1) immune checkpoint inhibition has been unsuccessful in clinical trials. Myeloid-derived suppressor cells (MDSCs) are known to block anti-tumor CD8+ T cell immune responses in various cancers including pancreas. This has led us to our objective that was to develop a clinically relevant in vitro organoid model to specifically target mechanisms that deplete MDSCs as a therapeutic strategy for PDAC. *Method*: Murine and human pancreatic ductal adenocarcinoma (PDAC) autologous organoid/immune cell co-cultures were used to test whether PDAC can be effectively treated with combinatorial therapy involving PD-1 inhibition and MDSC depletion. *Results*: Murine in vivo orthotopic and in vitro organoid/immune cell co-culture models demonstrated that polymorphonuclear (PMN)-MDSCs promoted tumor growth and suppressed cytotoxic T lymphocyte (CTL) proliferation, leading to diminished efficacy of checkpoint inhibition. Mouse- and human-derived organoid/immune cell co-cultures revealed that PD-L1-expressing organoids were unresponsive to nivolumab in vitro in the presence of PMN-MDSCs. Depletion of arginase 1-expressing PMN-MDSCs within these co-cultures rendered the organoids susceptible to anti-PD-1/PD-L1-induced cancer cell death. *Conclusions*: Here we use mouse- and human-derived autologous pancreatic cancer organoid/immune cell co-cultures to demonstrate that elevated infiltration of polymorphonuclear (PMN)-MDSCs within the PDAC tumor microenvironment inhibit T cell effector function, regardless of PD-1/PD-L1 inhibition. We present a pre-clinical model that may predict the efficacy of targeted therapies to improve the outcome of patients with this aggressive and otherwise unpredictable malignancy.

## 1. Introduction

Pancreatic cancer is currently the third most common cause of cancer-related death in the United States [1,2]. Pancreatic ductal adenocarcinoma (PDAC) is one of the most lethal malignancies, with an approximate 10% five-year survival rate despite therapy [1,2]. The dismal response of PDAC to various existing treatments contributes to its poor prognosis and makes this disease an unmet medical challenge.

Tumors can evade immune surveillance by expressing immune checkpoint molecules. For example, programmed cell death 1 ligand (PD-L1) interacts with the T cell protein programmed death 1 (PD-1), subsequently inhibiting CD8+ cytotoxic T lymphocyte proliferation, survival and effector function [3,4,5]. According to The Cancer Genome Atlas (TCGA), data using CBioPortal analysis website results showed that approximately 33% PDAC patients express PD-L1 [6]. Although anti-PD-1 antibodies have already been tested in clinical trials for pancreatic cancer treatment, patients have failed to respond. A plausible reason for this observation may be that other redundant, immune-suppressive mechanisms are at play. Multiple suppressive immune cell types including macrophages, myeloid-derived suppressor cells (MDSCs) and regulatory T cells (Tregs) accumulate in early pancreatic intraepithelial neoplasia (PanIN), the precursor lesion to pancreatic cancer, and persist through cancer progression [7]. MDSCs are known to block CD8+ T cell anti-tumor activity through L-arginine and L-cysteine sequestration as well as reactive oxygen species (ROS) [8,9,10]. In support of these findings, pre-clinical mouse models have similarly demonstrated that targeting the MDSC population enables enhanced endogenous T cell response in PDAC [11]. In addition, in a study of circulating and splenic immune cells, PDAC patients that exhibited reduced levels of CD8+ T cells also exhibited elevated levels of MDSCs [12]. In humans and mice, two distinct subsets of MDSCs exist—monocytic-MDSCs (M-MDSCs) and polymorphonuclear-MDSCs (PMN-MDSCs) [9,10]. PMN-MDSCs are known to be associated with a poor prognosis in pancreatic cancer [11]. 

We have generated an organoid/immune cell-based pre-clinical model for the study of PMN-MDSCs within the tumor microenvironment that deviates from the existing co-cultures reported [13,14,15]. In particular, we find that PD-L1-positive pancreatic cancer cells can be effectively treated with a combinatorial treatment involving the inhibition of PD-L1/PD-1 interaction and PMN-MDSC depletion.

## 2. Results

### 2.1. Increased PMN-MDCS Infiltration Correlated with Tumor Growth in Orthotopically Transplanted Mice

Mice were orthotopically transplanted with syngeneic 7940B PDAC cells derived from a primary spontaneous PDAC tumor arising in the body of the pancreas (C57BL/6) of a male transgenic *Kras*^LSL-G12D/+^, *Trp53*^LSL-R172H/+^, *Pdx1-Cre* (KPC) mouse [16]. Seven days post-transplantation, mice were treated with chemotherapeutics (gemcitabine plus paclitaxel-like Epothilone A), *InVivo*Plus anti-mouse PD-1 (CD279), cabozantinib (kinase inhibitor known to deplete MDSCs) or a combination of two or three of these drugs for a further 7 days post-orthotopic transplantation. The flow cytometric gating strategy for EpCAM^+^/PD-L1^+^, CD8+ and PMN-MDSCs is shown in Appendix A. The percentages of these cell populations in untreated tumor tissue are quantified in Appendix A. Mice treated with a combination of anti-PD-1 and cabozantinib or triple treatment (chemotherapy, anti-PD-1 and cabozantinib) exhibited significantly decreased tumor weights (Figure 1A), increased EpCAM/PD-L1-expressing cell death (Figure 1B) and increased PMN-MDSC death (Figure 1C). Decreased tumor weights in the combination-treated mice correlated with increased CTL proliferation within the tumor tissue (Figure 1D). The overall survival was significantly increased in mice treated with either a combination of anti-PD-1 and cabozantinib or triple treatment as measured by a Kaplan–Meier curve (Appendix A). 

Digital spatial profiling (DSP) of immune-related protein markers across experimental mouse groups and PDAC patient tissue samples was performed. Regions of interest (ROIs) were selected based on tumor (PanCK), immune (CD8, CD45), and stromal (SMA high) regions (Appendix A). DSP demonstrated that in areas of decreased stromal cell infiltration in response to cabo+PD-1Inh treatment, there was an infiltration of CD8+GZMB+Ki67+ cytotoxic T lymphocytes and decreased immunosuppressive immune cell populations (Figure 1E,F). In particular, compared to controls, mice treated with either cabo+PD-1Inh, or chemo+cabo+PD-1Inh triple treatment, significantly decreased tumor weights correlated with a significant decrease in stromal markers alpha smooth muscle actin (Figure 1I), fibronectin and vimentin, and PMN-MDSCs (Figure 1F,H) infiltration, with an increase in CD8+ infiltrating cells (Figure 1E–G). In support of these observations, quantitative RT-PCR confirmed the observations made by the DSP analyses in that a significant increase in CD8 (Figure 1J) and granzyme (Figure 1K) expression in tumors collected from cabo+PD-1Inh, or chemo+cabo+PD-1Inh triple-treated mice, correlated with a decrease in fibronectin expression (Figure 1L).

To validate the findings that decreased tumor weights have a strong negative correlation with CTL proliferation (CD8+/BrdU+ Cells, Figure 1A,D) in the combination-treated mice, a regression analysis was performed between the two variables (tumor/body weight vs. cell proliferation, Appendix A) of all experimental groups. The data strongly support the finding that combination-treated mice possessed an increased number of CD8+/BrdU+ cells (90, )Appendix A) with a decreased tumor mass (900 mg) compared to their untreated control (Appendix A). The summarized column-line graph (Appendix A) clearly showed the inverse relationship between tumor weight and CTL proliferation.

### 2.2. Organoids Derived from Cabozantinib-Treated Mouse Tumors Exhibit a Decreased Stromal Cell Compartment That Correlates with Increased CD8+ Cells

Organoids were derived from tumor tissues collected from the eight experimental groups shown in Figure 1. Light micrographs of organoids in culture (Figure 2A) and H&E stains of embedded organoids (Figure 2B) demonstrated morphological changes and decreased efficiency of growth in cultures derived from cabo+PD-1Inh, and chemo+cabo+PD-1Inh-treated mice. Cultures were then directly analyzed by flow cytometry for PMN-MDSCs, CD8+ and SMA+ cells carried forward from tumor tissues into the organoid cultures (Figure 2). Organoids derived from mouse groups treated with cabozantinib showed with a significant decrease in PMN-MDSCs reflective of decreased cell viability (Figure 2C,E). The decrease in PMN-MDSCs correlated with a significant increase in CD8+ cells in cultures derived from cabo+PD-1Inh and chemo+cabo+PD-1Inh-treated mice (Figure 2D,E). An increase in CD8+ cells that were carried forward from tumor tissues to organoid cultures, correlated with a significant decrease in SMA-positive cells (Figure 2D,E). Overall, cabozantinib treatment resulted in a decrease in the number of SMA-positive cells observed in organoid cultures (Figure 2D,E).

Collectively, our in vivo and in vitro studies in the PDAC orthotopic mouse and organoid models demonstrate that PMN-MDSCs are likely to contribute to tumor growth, suppression of CD8+ T cell proliferation and effector function that may lead to disruption of the efficacy of checkpoint inhibition. We also documented a significant reduction in the stroma, both in vivo and in vitro, in response to cabozantinib treatment.

### 2.3. PMN-MDCSs Disrupt the Efficacy of Checkpoint Inhibition in Mouse-Derived Organoid/Immune Cell Co-Cultures

To investigate whether PMN-MDSCs disrupt the efficacy of checkpoint inhibition in PDAC tumor survival, we developed a pancreatic cancer organoid/CTL/MDSC co-culture. Figure 3A is an overview of the experimental approach developed by the research team to co-culture pancreatic cancer organoids with autologous immune cells. The protocol is executed, and data analyzed within 10 days of the start of organoid and immune cell cultures (Figure 3A). Importantly, tumor antigen-pulsing of DCs and CTL activation at day 4 of the protocol is fundamental and attempts to develop a system that is closest to physiological relevance. Mouse-derived pancreatic cancer organoids secreted cytokines including IL-10, IL-6, GM-CSF, and growth factors VEGF and TGFβ, which are all intimately involved in the development of cells of the myeloid cell lineage (Figure 3B). Normal mouse-derived pancreatic organoids showed no expression of secreted cytokines and growth factors (Figure 3C). Mice were orthotopically transplanted with syngeneic 7940B pancreatic cancer cells. After 14 days, pancreatic cancer tumors were collected. Bone marrow and splenocytes were extracted from mice. Monocytes and dendritic cells were derived from bone marrow and CTLs were sorted from splenocytes. Autologous organoid/CTL co-cultures treated with *InVivo*Plus anti-mouse PD-1 (PD-1Inh) exhibited significant organoid death (condition 2, Figure 3D,E,I) compared to untreated controls (condition 1, Figure 3D,F,J). Importantly, when MDSCs were added to the co-culture this response was inhibited (condition 3, Figure 3D,G,K). The addition of cabozantinib (cabo) to the organoid/CTL/MDSC co-culture depleted MDSCs from culture and maximized the efficacy of checkpoint inhibition to induce PD-L1-expressing cancer organoid death (condition 4, Figure 3D,H,L). Organoid death was quantified by flow cytometry of zombie (viability dye)^+^/EpCAM^+^/PD-L1^+^ cells (Figure 4A).

To identify the impact of PMN-MDSCs on CD8+ T cell proliferation, we next assayed cell CFSE uptake within the same co-culture. CD8+ T cells within cultures of organoids without autologous PMN-MDSCs exhibited a significant increase in CTL proliferation in response to PD-1Inh (condition 2, Figure 4D). This proliferative response was blocked with the introduction of PMN-MDSCs within the co-culture (condition 3, Figure 4D). Combinatorial treatment with PD-1Inh and cabozantinib resulted in the induction of CD8+ T cell proliferation (condition 4, Figure 4C,D) compared to condition 1 (Figure 4C,D). Decreased CD8+ T cell proliferation in response to the introduction of PMN-MDSCs into the co-culture correlated with a significant decrease in IFNγ and IL-2 expression in CTLs within the same culture (Figure 4B). 

Quantitative RT-PCR was performed using RNA extracted from magnetically separated EpCAM+, MDSC+ and CD8+ cell fractions from within the co-culture (Figure 4E). CD8+ expression within cultures of organoids without autologous PMN-MDSCs exhibited a significant increase in CTL proliferation in response to PD-1Inh (condition 2, Figure 4G). CD8 expression significantly decreased with the introduction of PMN-MDSCs within the co-cultures (condition 3, Figure 4G) and correlated with increased Arg1, NOS2 and CD11b expression (Figure 4F). Treatment with PD-1Inh alone or in combination with cabozantinib, resulted in the induction of CD8+ granzyme B expression and decreased PD-L1 and EpCAM expression (condition 4, Figure 4H). Appendix A shows a significant inhibitory effect on CTL proliferation with increased numbers of PMN-MDSCs within co-cultures.

### 2.4. Generation of PDAC Patient-Derived Organoids and Orthotopic Transplantation

Figure 5A,B are representative of one normal and one tumor organoid line, the latter derived from a pancreatic cancer patient with stage 3 cancer, which display morphological differences. While organoids derived from normal pancreatic tissue exhibit a spherical morphology lined with ductal epithelium (Figure 5A), PDAC organoids appeared lobular and irregular (Figure 5B). Both normal and tumor human-derived pancreatic organoids expressed lineage markers of the pancreas including HNF-1β, CK19 and SOX9 (Figure 5C,D). Organoids that were PD-L1 positive were used for subsequent co-culture experiments. Appendix A is a representative immunofluorescence stain demonstrating the divergence in morphology between individual organoid line and the expression of PD-L1 and stem cell marker CD44v9. Information regarding treatment, tumor response and stage for patients from which the organoid lines were derived is detailed in Appendix A. Orthotopic transplantation of the human-derived PDAC organoids into NOD scid gamma (NSG) mice resulted in the development of PDAC 4 weeks post-engraftment (Figure 5E–O). Importantly, mice transplanted with these organoids exhibited lesions in the spleen and peritoneum of the same mouse mimicking early stage metastasis in human patients as confirmed by a board-certified pathologist (Figure 5G–O). We confirmed that these lesions were derived from the patient’s organoids by using a human-specific antibody against histone (Figure 5I,L,O).

Using human-specific antibodies, KRAS mutation and PD-L1 expression were detected in the patient tumor tissue, corresponding patient-derived organoids (Figure 5P). Pancreatic tumor tissue collected from NSG mice orthotopically transplanted with patient-derived organoids also expressed KRAS mutation and PD-L1 consistent with the patient’s organoid culture and tumor tissue (Figure 5P). 

### 2.5. DSP Reveals a Correlation between Infiltrating PMN-MDSCs, Arg1 Expression and Increased Stroma in Patient Tissue

Tumor, stromal and immune ROIs of PDAC were identified on a pancreatic tumor tissue array by PanCK, SMA and CD68 staining respectively (Figure 6A). Figure 6B shows quantitative expression of tumor, stromal and immune cells in the tissue array. While a Pearson correlation matrix (Figure 6C) of all proteins measured allowed us to identify a significant correlation between increased MDSCs and PD-L1 (Figure 6C,D), a significant inverse correlation was observed between CD8 and markers specific for the stroma (Figure 6C,E). Among the proteins that were significantly correlated with the stromal markers were Arg1 (Figure 6F–H), and S100B (Figure 6I–K).

Patient-derived PDAC organoids secreted cytokines including IL-1β, IL-6, GM-CSF, and VEGF (Appendix A). While cabozantinib significantly induced decreased Arg1 and NOS2 expression in PMN-MDSCs, sunitinib and regorafenib had no effect (Appendix A). PMN-MDSCs were differentiated based on our protocol and co-cultured with tumor antigen-pulsed DC-activated CTLs. Using a flow cytometric-based CFSE T cell proliferation assay, we observed that sunitinib (STAT3 signaling pathway inhibitor) decreased MDSC-mediated immunosuppression of CTLs (Appendix A) almost as effectively as cabozantinib (Appendix A). While regorafenib (inhibitor of the MAPK signaling pathway) had no effect on PMN-MDSC-mediated suppression of CTL proliferation (Appendix A) when compared to vehicle (Appendix A). Such findings further support an immunosuppressive role of PMN-MDSCs.

### 2.6. Depletion of PMN-MDSCs from Patient-Derived Organoid/Immune Cell Co-Cultures Maximizes the Effect of Anti-PD-1/PD-L1 Interaction

We established a PDAC patient-derived autologous organoid/immune cell co-culture in which CTLs were activated following a 72 h co-culture with autologous dendritic cells that had been pulsed with the conditioned media from autologous patient-derived PDAC organoids (Figure 7A). Following this co-culture CTLs were extracted from dendritic cells and co-cultured with either autologous patient-derived PDAC organoids alone or with autologous PMN-MDSCs. Immunofluorescence showed the direct contact between CTLs and PDAC organoids (Figure 7A). Figure 7B is a schematic diagram summarizing the composition and treatments of co-culture conditions 1–5. Flow cytometric CFSE assay clearly demonstrated an increase in CTL proliferation in response to nivolumab (PD-1Inh) treatment (condition 2) compared to the no treatment group (condition 1, Figure 7B–D). Cabozantinib alone had no effect on CTL proliferation (condition 3, Figure 7B–D). When PMN-MDSCs were co-cultured with organoids and CTLs (condition 4) (Figure 7B), nivolumab-induced CTL proliferation was significantly inhibited (Figure 7C,D). Depletion of PMN-MDSCs from the co-cultures resulted in a significant increase in CTL proliferation (condition 5, Figure 7B–D). Increased CTL proliferation in treatment groups 2 and 5, correlated with a significant increase in perforin as measured by flow cytometry in the same cultures (Figure 7E).

In an autologous organoid/CTL co-culture derived from patients with PD-L1-positive tumors, organoids were sensitive to nivolumab treatment (condition 2), whereby the addition of PMN-MDSCs inhibited nivolumab-induced EpCAM^+^ cancer organoid death (condition 4, Figure 7F). Depleting the co-cultures of PMN-MDSCs with the addition of cabozantinib maximized the efficacy of checkpoint inhibition and significantly increased EpCAM^+^-expressing organoid cell death (condition 5, Figure 7F). Cabozantinib alone had no effect on the viability of EpCAM^+^ cancer cells (condition 3, Figure 7F). Flow cytometric analysis demonstrated a significant decrease in PMN-MDSCs (CD33^+^CD14^−^CD11b^+^CD15^+^Arg1^+^) in response to cabozantinib treatment within co-cultures (condition 5, Figure 7B,G), when compared to condition 4 (Figure 7B,G). Collectively, these data suggest that a combinatorial treatment that depletes PMN-MDSCs and inhibits the interaction between PD-1/PD-L1 is necessary to enhance CTL effector function and target PD-L1-expressing PDAC cells.

## 3. Discussion

Using mouse- and human-derived autologous pancreatic cancer organoid/immune cell co-cultures, we demonstrate that elevated infiltration of polymorphonuclear (PMN)-MDSCs within the PDAC tumor microenvironment inhibits T cell effector function, regardless of PD-1/PD-L1 inhibition. Anti-PD-1 antibodies have been tested in clinical trials for pancreatic cancer treatment, but patients have failed to respond [17]. Our studies demonstrate that PMN-MDSCs contribute to the immune-suppressive mechanisms that render responses to immunotherapy ineffective. Previous studies have clearly shown that the depletion of myeloid cells impairs tumor growth at different stages of pancreatic carcinogenesis in vivo [8,11,18], and PMN-MDSCs inhibit CTL anti-tumor activity through L-arginine and L-cysteine sequestration as well as production of ROS [8,9,10] (Figure 8). Indeed, we observed both in vivo and in vitro that depletion of PMN-MDSCs using cabozantinib resulted in the sensitization of cancer cells to anti-PD-1/PD-L1 treatment that correlated with significant decreases in NOS2 and Arg1 and increased CTL proliferation and effector function. PMN-MDSCs are known to be associated with a poor prognosis in pancreatic cancer [11], yet pre-clinical studies that evaluate the efficacy of targeting MDSCs in combination with immunotherapy have not been performed.

Depletion of PMN-MDSC using cabozantinib resulted in increased CTL proliferation and effector function both in vivo and in vitro. The first report of targeted depletion of MDSCs to unmask endogenous T cell response in PDAC was by Stromnes et al. [11]. In another study using a genetically engineered mouse model that endogenously expressed a single mutant *Kras* allele in progenitor cells of the pancreas, investigators demonstrated that depletion of granulocytic (polymorphonuclear)-MDSCs using a monoclonal antibody (*InVivo*Plus anti-mouse Ly6G, Bioxcell), resulted in increased accumulation of activated CD8+ T cells and apoptosis of epithelial cells [7,11]. In an orthotopically transplanted mouse model using pancreatic cancer cells isolated from tumors of transgenic *Kras*^LSL-G12D/+^, *Trp53*^LSL-R172H/+^, *Pdx1-Cre* (KPC) mice that express PD-L1, tumors were sensitized to anti-PD-1/PD-L1 therapy in response to cabozantinib-depleted PMN-MDSCs. The studies by Stromnes et al. [11] elegantly demonstrate the use of a specific antibody for the targeted depletion of PMN (granulocytic)-MDSCs. While we acknowledge that cabozantinib may also target other cells besides the PMN-MDSCs, our rationale for using this drug was based on pre-clinical data and clinical trials in a number of other cancers. For example, evidence from prostate, renal and breast cancer research, and clinical trials, has clearly documented that tyrosine kinase inhibitors (such as cabozantinib and sunitinib) target MDSC function or generation [19,20,21]. In fact, cabozantinib-targeted MDSC depletion used in combination with immune checkpoint blockade induces anti-tumor activities and tumor regression in castration-resistant prostate cancer [21]. These studies prompted us to hypothesize that PDAC could potentially be effectively treated via the combined action of immune checkpoint blockade and targeted depletion/inactivation of MDSCs (Figure 8). Based on the literature, we know that cabozantinib is a tyrosine kinase inhibitor that targets MET, VEGFR2, FLT3, c-KIT and RET [22]. In addition, sunitinib has been shown to be effective in inducing renal tumor cell apoptosis by targeting the STAT3 pathway [23]. These studies are of importance given that our data show that mouse- and patient-derived pancreatic cancer organoids secrete cytokines including IL-6, GM-CSF, and growth factor VEGF, which are all intimately involved in the development of cells of the myeloid cell lineage (Figure 8). Both cabozantinib and sunitinib effectively blocked the immunosuppressive function of MDSCs on CTL proliferation. However, cabozantinib alone resulted in decreased Arg1 and NOS2 expression in MDSCs. These data may suggest independent signaling pathways regulating MDSC differentiation and ROS production, but the exact mechanism by which cabozantinib acts on PMN-MDSCs is still unclear in our system and warrants further investigation.

Digital spatial profiling of PDAC patient tissues demonstrated that there was a significant infiltration of CD68-positive monocyte immune cells that highly expressed arginase 1 (Arg1), CD66b, V-domain Ig suppressor of T cell activation (VISTA) and indoleamine 2,3-dioxygease (IDO1). The protein expression pattern of these CD68-positive immune cells was consistent to the PMN-MDSC phenotype. In support of our findings, MDSCs utilize a number of mechanisms to suppress anti-tumor immunity. Such mechanisms include the expression of high levels of Arg1, iNOS or ROS, as well as the production of IDO1 [24,25,26]. The IDO and kynurenic pathway (L-tryptophan, L-kynurenine) also blocks apoptosis in pancreatic cancer cells and promotes disease progression [27]. In addition, evidence from acute myeloid leukemia (AML) demonstrates a strong positive association between MDSC expression of VISTA and T cell expression of PD-1 [28]. Recent characterization of the infiltrating PMN-MDSCs in response to *Helicobacter* infection in the stomach showed that these immunosuppressive cells have a unique signature. These Arg1+/NOS2+ PMN-MDSC population express the Schlafen (SLFN) family of proteins, a transcriptional target of GLI1, and the endogenous small, non-coding RNA, MiR130b [29,30]. SLFN12L^+^ myeloid cells express VEGF, IL-1β and TNF-α, which are known factors associated with MDSC regulation, an immunosuppressive cell of relevance to immunotherapy-resistant gastric cancer [29,30]. Thus, we may speculate that these unique PMN-MDSCs are sustained within the gastric TME. More recently, a phase IIa, open-label, two-cohort study assessed the efficacy of the CXCR4 antagonist BL-8040 (motixafortide) with pembrolizumab and chemotherapy in metastatic PDAC [31]. This study demonstrated that BL-8040 increased CD8+ effector T cell tumor infiltration and decreased myeloid-derived suppressor cells (MDSCs), as well as expanding the beneficial effect of chemotherapy in PDAC patients [31]. Collectively, these unique features of MDSCs provide insight into their immunosuppressive role in clinical disease. Thus, the depletion of MDSCs to impair tumor growth may be proposed [32] (Figure 8).

The inhibitory effect of cabozantinib on the stromal cell compartment within the tumors of treated mice (Figure 1) was unexpected. Targeting the stroma to effectively treat pancreatic cancer is aimed to increase delivery of standard-of-care agents to tumor cells, to improve immune surveillance and to inhibit tumor-promoting signaling from cancer associated fibroblasts [33]. Most clinical trials using therapies targeting the stroma have so far shown limited benefits for patients with advanced stage disease [34,35]. There are currently no reports of the effect of combinatorial therapy using tyrosine kinase inhibitors with anti-PD-1/PD-L1 immunotherapy for the treatment of pancreatic cancer. We also noted a strong correlation between stromal markers and S100B in patient PDAC tissues. Elevated S100B is of relevance to our studies, given that S100 proteins are known to regulate the accumulation of MDSCs by binding cell surface glycoprotein receptors on MDSCs, signal via the NF-*κ*B pathway, and promote MDSC migration [36].

Clinically relevant approaches for the effective treatment of PDAC that specifically target mechanisms of MDSC depletion are unclear. This has led us to our rationale for the development of the current co-culture system that may be used to identify such targeted MDSC disruption. Indeed, other organoid-based patient profiling efforts have been developed, including co-culture models. Our approach may be considered closer to what occurs within the tumor microenvironment. Pancreatic cancer organoids have been co-cultured with immune cells outside of the Matrigel dome [13]. This is a non-autologous system with immune cells placed outside of the Matrigel dome and not allowing the immune cells to ‘roll’ towards target organoids. Similar methods have been used in autologous non-small-cell lung and colorectal cancer organoid/peripheral blood lymphocyte co-cultures [14]. In contrast, CD8+ T cells in the co-culture system reported here, are activated using the patient’s own tumor antigen and dendritic antigen presenting cells. Additionally, PMN-MDSCs are differentiated using conditioned media collected from the patient’s own organoid line. Our in vitro differentiation of the PMN-MDSCs may recapitulate the in vivo mechanism whereby tumor antigen-conditioned MDSCs impede the acquisition of CTL effector function. Mouse (Figure 1) and human (Figure 6) data presented here demonstrate that, given the dense stromal compartment of the PDAC microenvironment, isolation of immune cells from native tumor tissue to carry forward in the organoids may not be feasible. A number of mechanisms may be targeted to deplete the PMN-MDSCs within the tumor microenvironment of the pancreas including their differentiation, trafficking, maturation and immunosuppressive function. Such mechanisms may specifically be interrogated in an organoid/immune cell co-culture system such as that reported here.

Retrospective evaluation of patient response is the critical first step for use of organoid technology for personalized PDAC treatment. If such studies were successful, then a prospective study could follow with the goal of identifying and prioritizing treatment options to directly improve patient survival. Genomic, epigenomic and transcriptomic analyses are fundamental for the characterization of molecular subtypes of pancreatic cancer that can define candidate molecular mechanisms in the tumor microenvironment. For example, the immunogenic subtype of pancreatic cancer identified by Bailey et al. clearly demonstrated the predominant expression of profiles related to increased B, CD8+ and regulatory T cell infiltration, and increased acquired tumor immune suppression pathways [37]. While biological insights can be gained from such genomic studies, organoid cultures need to be used in parallel to faithfully represent the therapeutic sensitivity profile according to the genetic heterogeneity of the primary tumor. The study by Tiriac et al. is a demonstration of the use of patient-derived organoid profiling using next-generation sequencing of DNA and RNA complemented with pharmacotyping to predict patient response [38]. However, a challenge of such studies is the lack of the immune compartment in the organoid culture. The primary goal of our studies is to leverage the patient-derived organoid/immune cell co-cultures to determine response to therapy in clinical trials. Based on our pre-clinical data, a developed trial using a combination of cabozantinib and atezolizumab will potentially increase response in patients with metastatic PDAC. This trial will be run in parallel with correlative studies using the organoid/immune cell co-culture technology to understand and define the changes in the patient’s tumor landscape with combinatorial therapy. For future studies, it is critical to refine ex vivo organoid co-culturing approaches which faithfully represent the patient’s tumor microenvironment.

## 4. Methods

### 4.1. Mouse Orthotopic Transplants and Treatment

All mouse studies were approved by the University of Cincinnati Institutional Animal Care and Use Committee (IACUC) (protocol: 07-01-11-01, approval date: 17 September 2018) that maintains an American Association of Assessment and Accreditation of Laboratory Animal Care (AAALAC) facility. C57BL/6 mice purchased from Jackson Laboratories were orthotopically transplanted with 500,000 7940B cells derived from a primary spontaneous PDAC tumor arising in the body of the pancreas (C57BL/6) of a male transgenic *Kras*^LSL-G12D/+^, *Trp53*^LSL-R172H/+^, *Pdx1-Cre* (KPC) mouse [39,40] (kindly donated by Dr. Gregory Beatty, University of Pennsylvania). Nod scid gamma mice were orthotopically transplanted with 500–1000 pancreatic cancer organoids derived from PDAC patients.

In a separate series of experiments, 7 days post-orthotopic transplantation, C57BL/6 mice (Jackson Laboratories) were treated with gemcitabine (Selleckchem, S1149) (325 μg/mouse, i.p.) every 2 weekdays and Epothilone A (abaraxane, Selleckchem, S1297) (13 μg/mouse, i.v.) every 5 days, *InVivoPlus* anti-mouse PD-1 (BioXCell, BP0146) (200 μg/mouse, i.p.) every 5 days, Cabozantinib (cabo, Selleckchem, S1119) (780 μg/mouse, oral gavage) every weekday or a combination of 3 or 4 drugs for 7 days. The mice were sacrificed, tumors were removed and weighed.

### 4.2. Generation of Mouse- and Human-Derived Pancreatic Cancer Organoids

Tumor tissue was obtained from patients undergoing surgical resection for pancreatic cancer (IRB protocol number: 2015-5537, University of Cincinnati). Pancreatic biopsies were obtained from patients during esophagogastroduodenoscopy using ultrasound-guided fine needle aspiration (IRB protocol number: 1909985869R001, Tissue Acquisition and Repository for Gastrointestinal and HEpaTic Systems, University of Arizona). Mouse and human pancreatic tumors were harvested in wash buffer (DBPS, Corning, 21-031-CV) supplemented with 1% Penicillin/Streptomycin (Corning, 30-002-CI), 1% Kanamycin Sulfate (ThermoFisher Scientific, Waltham, MA, USA, 15160-054) and 0.2% Gentamicin/Amphotericin B (ThermoFisher Scientific, S1714). Tumor tissue was then mechanically minced with razor blades and suspended in 5 mL of HBSS (Corning, 21-021-CV) supplemented with 5% FCS (Atlanta Biologicals, S12450H), 1% Penicillin/Streptomycin (Corning, 30-002-CI), and 1% Kanamycin Sulfate (ThermoFisher Scientific, 15160-054) containing 1 mg/mL Collagenase P (Roche, 11213865001). Tumor tissue was digested by incubating at 37 °C for 15 min. After 15 min, tissue fragments were analyzed under a microscope. If clusters of cells were present, the reaction was stopped with 5 mL of HBSS supplemented with 5% FCS (Atlanta Biologicals, S12450H), 1% Penicillin/Streptomycin, 1% Kanamycin Sulfate and 0.1% Gentamicin/Amphotericin B. Digested tissue was rinsed in wash buffer and filtered through a 70 μM filter.

Filtrate was carefully removed and centrifuged at 400× *g* for 5 min. Cells were washed again with DPBS wash buffer and then suspended in Matrigel^TM^ (Fisher Scientific, CB40230C) supplemented with 1% Penicillin/Streptomycin and then overlaid with organoid media. Human-derived organoids (HuTPOs) were cultured in human pancreatic media (advanced DMEM/F12 (Thermo Fisher Scientific, 1263-010), 1× B27 (Thermo Fisher Scientific, 12587010), 284 μM ascorbic acid (R&D Systems, 4055-50), 20 μg/μL Insulin (R&D, 3435/10), 0.25 μg/μL hydrocortisone (Sigma-Aldrich, H0888-1G), 100 ng/mL fibroblast growth factor-basic (FGF2, Peprotech, 100-18B), 100 nM Retinoic Acid (ATRA, Sigma, R2625), 10 μM Y27632 (Sigma-Aldrich, Y0503), 100 ng/mL fibroblast growth factor 10 (FGF10, Peprotech, 100-26), 1% Penicillin/Streptomycin, 0.1% Gentamicin/Amphotericin B, 2 mM GlutaMAX™ (Fisher Scientific, 350-50-061) and 56 μg/mL bovine pituitary extract (BPE, Sigma-Aldrich, P1476, 10% R-Spondin, and 50% Wnt conditioned media). Mouse-derived organoids were cultured in modified mouse pancreatic media (advanced DMEM/F12, 1% Penicillin/Streptomycin, 1× B27 (Thermo Fisher Scientific, 12587010), 1.25 mM N-acetyl cysteine (Sigma-Aldrich, A7250), 10 nM gastrin (Tocris, 3006/1), 50 ng/mL epidermal growth factor (EGF, Peprotech, 315-09), 10% R-Spondin, 100 ng/mL Noggin (Peprotech, 250-38), 100 ng/mL FGF10 (Peprotech, 100-26), 10 mM nicotinamide (Sigma-Aldrich, N0636), 10% R-Spondin, and 50% Wnt conditioned media [41].

An alternative method was obtained to generate human-derived pancreatic cancer organoids, while there was abundance of fat globules. After the digestion reaction was stopped with HBSS supplemented with 5% FCS, 1% Penicillin/Streptomycin, 1% Kanamycin Sulfate, and 0.1% Gentamicin/Amphotericin B, the digested tissue was filtered through a 500 micron and 100 micron sterile metal mesh, simultaneously. A volume of 10 mL of pancreatic filtration media (HBSS supplemented with 30% FCS, 1% Penicillin/Streptomycin) in a 50 mL conical was vortexed until a 1 inch layer of foam was formed. Using a serological pipet, the flow through from the 100 micron filter was passed through the foam slowly in order to trap the fat globules in the foam. The cells were allowed to settle for 1 min and the liquid from bottom was collected carefully without disturbing the foam layer and transferred to a 15 mL conical tube. Cells were centrifuged at 400× *g* for 5 min. The cell pellet was resuspended in DPBS wash buffer and centrifuged at 4 °C at 400× *g* for 5 min. Cells were embedded in Matrigel^TM^ and then overlaid with organoid media.

### 4.3. Extraction and Culture of Murine and Human Immune Cells

Murine monocytes were isolated and cultured from bone marrow according to a published protocol [42,43]. Dendritic cells were cultured from bone marrow-derived monocytes according to [43]. CTLs were extracted from splenocytes using the EasySep^TM^ Mouse CD8+ T cell Isolation kit according to the manufacturer’s protocol (Stemcell Technologies, 19853) and cultured according to previously published studies [43].

Whole blood was collected from PDAC patient blood (IRB protocol number: 2015-5537, University of Cincinnati). Sepmate^TM^ tubes (Stemcell Technologies, 85450) containing Lymphoprep^TM^ (Stemcell Technologies, 01-63-12-001-A) were used to separate out red blood cells and platelets according to the manufacturer’s protocol. The resulting peripheral blood mononuclear cells (PBMCs) were either cultured in human monocyte media, human dendritic cell media or put through the negative selection EasySep^TM^ Human CD8+ T Cell Enrichment Kit (StemCell Technologies, 19053). PBMCs were cultured into dendritic cells using a modified published protocol consisting of AIMV media (Thermo Fisher Scientific, 12055091), containing, 10% human serum albumin (HSA, Gemini Bioscience, 800-120), 50 μM β-mercaptoethanol (Thermo Fisher Scientific, 21985023), 1% Penicillin/Streptomycin, 800 U/mL GM-CSF (Thermo Fisher Scientific, PHC6025), 500 U/mL IL-4 (Thermo Fisher Scientific, RIL4I). Dendritic cells were then matured 72 h after culture using a maturation media consisting of human dendritic cell medium supplemented with 10 ng/mL IL-1β (Thermo Fisher Scientific, RIL1BI), 10 ng/mL 1L-6 (Thermo Fisher Scientific, RIL6I), 1 μg/mL PGE2 (TOCRIS, 2296) and 10 ng/mL TNF-α (Thermo Fisher Scientific, PHC3015) Dendritic cells were kept in maturation media for 16 h. Human CD8+ T Cells extracted from PBMCs were cultured for 16 h using a protocol adapted from a published protocol [44] and media consisting of RPMI 1640 medium, 10% human serum albumin, 1% Penicillin/Streptomycin, 50 μM β-mercaptoethanol, 1× ITS (Thermo Fisher Scientific, 41400045), 0.15 μg/mL IL-2 (Thermofisher Scientific, RIL2I), and 0.05 ng/mL IL-7 (Thermo Fisher Scientific, RP-8645). MDSCs were cultured in 50% MDSC culture media (AIMV media containing 1% Penicillin/Streptomycin, 10 ng/mL IL-1β, 10 ng/mL 1L-6, 1 μg/mL PGE2), 2 ng/mL TGF-β1 (Thermo Fisher Scientific, 7754-BH-005/CF), 10 ng/mL TNF-α, 10 ng/mL VEGF (Thermo Fisher Scientific, RVEGFI), 10 ng/mL GM-CSF) and 50% conditioned medium collected from autologous organoid culture for 7 days, removing half of the media and replacing with fresh media/conditioned media every 48 h.

### 4.4. Human- and Mouse-Derived Pancreatic Cancer/Immune Cell Co-Cultures

(Autologous Tumor Organoid and Immune Cell Co-Cultures and Methods of Use as Predictive Models for Pancreatic Cancer Treatment, PCT/US2020/014925, UC Ref. 2019-071, Patent Pending).

After maturation, dendritic cells were pulsed with either mouse- or human-derived pancreatic cancer organoid tumor conditioned media for 2 h by replacing the maturation media with 50% organoid conditioned media. Pulsed dendritic cells were then co-cultured with murine- or human-derived CD8+ cytotoxic T lymphocytes (CTLs) at a ratio of 1:4 (DCs:CTLs) for 72 h in DC/CTL co-culture media (RPMI 1640, 10% human serum albumin, 1% Penicillin/Streptomycin). At 72 h after co-cultures, CTLs were harvested, labeled with CFSE (BioLegend, 423801) and used in co-culture with either human- or mouse-derived pancreatic cancer organoids and monocytes. Human T cells were extracted from the CD8+ T Cell/dendritic cell co-culture using the EasySep^TM^ Human CD8-Positive Cell Enrichment Kit (Stemcell Technologies, 19053) following the manufacture’s protocol. Murine T cells were extracted from the CD8+ T Cell/dendritic cell co-culture using the EasySep^TM^ Murine CD8-Positive Enrichment kit (Stemcell Technologies, 19853) following the manufacture’s protocol. Organoids were harvested from the Matrigel^TM^ using cold DMEM/F12 and centrifuged at 400× *g* for 5 min at 4 °C. Organoids and CTLs and/or MDSCs were resuspended in Matrigel^TM^. Human and mouse co-cultures were cultured in human and murine pancreatic media respectively [41]. The organoid/immune cell co-culture was treated 24 h after seeding with either PD-1 inhibitor (*InVivoPlus* anti-mouse PD-1 (BioXCell, BP0146), or 0.5 µg/mL Nivolumab (Selleckchem, A2002)) and, or 10 µM cabozantanib. At 72 h after the treatment, co-culture organoids, CTLs and MDSCs were analyzed by immunofluorescence, flow cytometry and qRT-PCR.

In a separate experiment, organoids were harvested separately from the organoid/CTL/MDSC co-culture using a biotinylated EpCAM antibody (Thermo Fisher Scientific, 13-5791-82) bound to magnetic beads and CELLectin Biotin Binder kit (Thermo Fisher Scientific, 11533D). EpCAM-positive organoids were harvested and analyzed by qRT PCR. From the EpCAM-negative fraction, CTLs were isolated from the MDSCs using the CD8-Positive Enrichment Kit. Both MDSCs and CTLs were also analyzed by qRT PCR.

### 4.5. Human- and Mouse- Derived MDSC-CTL Titration Assay

After maturation for 16 hrs, dendritic cells were pulsed with either mouse- or human-derived pancreatic cancer-derived organoid conditioned media for 2 h followed by co-culturing with CD8+ CTLs for 72 h. MDSCs were also cultured for 7 days in presence of 50% MDSC media and 50% autologous tumor-derived organoid conditioned media. CTLs were harvested from the DC/CTL co-culture, labeled with CFSE (BioLegend) and added to the MDSC culture at a 0:1, 1:1, 1:4, and 1:16 ratio. The MDSC/CTL co-cultures were maintained for 72 h and both MDSCs and CTLs from different cultures were analyzed by flow cytometry. 

### 4.6. Testing Efficacy of Different MDSC Inhibitors Using Human- and Mouse-Derived MDSC-CTL Co-Culture

The MDSC/CTL (4:1) co-cultures were established following previously mentioned protocol and treated with Cabozantinib, Sunitinib (Selleckchem, S7781), and Regorafenib (Selleckchem, S1178). CTLs were harvested from the co-culture after 72 h and analyzed by flow cytometry. MDSCs were analyzed with quantitative RT-PCR.

### 4.7. Immunofluorescence and Immunohistochemistry

Organoids were fixed in 3.7% formaldehyde, permeabilized with 0.5% Triton X-100 for 20 min at room temperature and blocked with 2% normal donkey serum (Jackson Immunology, 017-000-121) for 1 h at room temperature. Human-derived cultures were immunostained using antibodies specific for HNF-1β (Novus Biologicals, NBP1-89680, Rabbit, 1:50), Sox 9 (Novus Biologicals, NBP2-52943, mouse, 1:250) CK19 (R&D Systems, AF3506, sheep,1:80), PD-L1 (Novus Biologicals, 76769, rabbit, 1:100), CD8a (R&D Systems, MAB1509, mouse, 1:60), Histone (abcam, ab125027, 1:100) or CD44V9 (CosmoBio, LKG-M003, Rat, 1:1000). Pancreatic organoids were then stained with a 1:100 dilution of secondary antibodies (594 donkey anti-mouse and -647 donkey anti-rabbit and 488 donkey anti-rat) and counter stained with Hoechst 33,342 (Thermo Fisher Scientific, H1399, 10 µg/mL) for 1 h at room temperature. Organoids were visualized using the Zeiss LSM710 and LSM880.

Human pancreatic tumor tissue and embedded organoids were fixed in 4% paraformaldehyde, embedded in paraffin and, sectioned (5 microns). After deparaffinization and antigen retrieval (Antigen Unmasking Solution, Vector Laboratories, Burlingame, CA), endogenous peroxidase activity was blocked using 0.3% hydrogen peroxide/methanol for 20 min. Slides were then blocked with 20% goat serum (ImmPRESS^TM^ HRP Anti-Rabbit IgG reagent kit, Vector Laboratories, MP-7401) for 20 min at room temperature, and then incubated with a 1:100 dilution of an anti-Kras (G12D Mutant) (GeneTex, GTX132407, Rabbit) or anti-PD-L1 (Cell Signaling Technologies, 13684) antibody overnight at 4 °C. Sections were then incubated with anti-rabbit ImmPRESS Ig (Vector Laboratories, MP-7401) for 30 min at room temperature. The color was then developed with peroxidase substrate solution from the ImmPACT DAB Peroxidase (HRP) Substrate Kit (Vector Laboratories, SK-4105). The slides were mounted using Permount (Fisher Scientific, SP15-100) and imaged using Nikon TS2 Inverted Tissue Culture scope.

### 4.8. Flow Cytometry

Human and mouse organoid and immune cell co-cultures were harvested and dissociated using Accutase^®^ (Fisher Scientific, A11105-01). Mouse tumor tissue was dissociated to single cells using 5 mg/mL collagenase P. The tissues were vortexed and incubated in 37 °C shaker for 30 min. The digestion was stopped by adding an equal volume of pancreatic wash buffer to the tissue. Cells were filtered through 70 micron sterile filter and centrifuged 400× *g* for 5 min followed by labeling for flow cytometric analysis. Cells were washed with DPBS and suspended in 1:100 dilution of the zombie UV (BioLegend, 423108). The cells were incubated in zombie for 15 min at room temperature, followed by incubation with cell surface antibody cocktail for 30 min at 4 °C. The cells were then washed using cell staining buffer by centrifuging at 300× *g* for 5 min. The cells were fixed in Fix and Perm buffer (BD Bioscience, 554714) for 20 min at 4 °C, followed by wash buffer, incubation with intracellular antigen, Perforin (human, BioLegend, 308120) for 30 min at 4 °C. The following cell surface antibodies were used and diluted 1:100 in cell staining buffer BioLegend, 420201), anti-PD-L1 (human, BioLegend, 351006; mouse, BioLegend, 124312), anti-CD8 (human, BioLegend, 344744), CD33 (human, BioLegend, 303428), CD14 (human, BioLegend, 367144), CD15 (human, BioLegend, 301922), CD11b (human, BioLegend, 301308; mouse, BioLegend, 101208), HLA-DR (human, BioLegend, 307640), arginase 1 (human, BioLegend, 369704), and EpCam (human, BioLegend, 324226; mouse, BioLegend, 128016), LY6C (mouse, BioLegend), and LY6G (mouse, BioLegend, 127628).

CTLs were analyzed using antibodies specific for anti-CD8 (human, BioLegend, 300911; mouse, BioLegend, 100711), anti-IL2 (human, BioLegend 500327; mouse, BioLegend, 503825) and anti-IFN-γ (human, BioLegend 502537; mouse, BioLegend, 505807). Myeloid cells were harvested in accutase and centrifuged at 300 g for 5 min. Supernatant was discarded, and cells were suspended in 100 μL of a 1:1000 dilution of the zombie red cocktail (BioLegend, 423109) and incubated at room temperature for 20 min. Human MDSCs were stained using antibodies specific for CD33 (BioLegend, 303427), HLA-DR (BioLegend, 307609), CD16 (BioLegend, 302017), CD14 (BioLegend, 325627), CD11b (BioLegend, 301307) and CD66b (BioLegend, 305103). Mouse MDSCs were stained using antibodies specific for CD11b (BioLegend, 101208), Ly6G (BioLegend, 127628), Ly6C (BioLegend, 128016) and CD11c (BioLegend, 117306). Samples were resuspended in cell staining buffer and analyzed on the Cytek^®^ Aurora and data analyzed using FlowJo software.

Human and murine CTLs and MDSCs from MDSC/CTL titration assay were analyzed using antibodies specific for human and mouse anti-CD8, anti- CD11b, anti- IL-2, anti- IFN-g mouse anti-LY6C, and anti-LY6G, human and mouse, CFSE, human anti-granzyme, anti-CD14, anti-CD15, anti-CD33, and anti-HLA-DR. In order to analyze the changes in PMN-MDSC, CD8 and SMA compartments in mouse tumor tissue and organoids, the following flow antibodies were used, CD8, IFNγ, IL-2, CD11b, EpCAM, LY6C, Ly6G, and SMA. 

Mouse tumor tissues from different treatment groups were washed with wash buffer (DPBS with antibiotics) and minced with razor blades into smaller pieces. The tissue fragments were transferred to 15 mL tube containing 5 mL pancreatic wash buffer and 5 mg collagenase P. The tissues were vortexed and incubated in 37 °C shaker for 30 min. The appearance of single cells or very small clusters was visualized under microscope. The digestion was stopped by adding equal volume of pancreatic wash buffer to the tissue. The cells were filtered through a 70 micron sterile filter and centrifuged 400× *g* for 5 min. Cells were washed with DPBS and the pellet was suspended in 1:100 dilution of the zombie UV and incubated in zombie for 15 min at room temperature, followed by incubation with cell surface antibody cocktail for 30 min at 4 °C. The cells were fixed with Fix and Perm buffer for 20 min at 4 °C, followed by washing with wash buffer and resuspended in cell staining buffer. The stained cells were analyzed by flow cytometry in BD LSRII and the data analysis was performed using FlowJo software.

### 4.9. Quantitative RT-PCR (qRT-PCR)

RNA was extracted from either tissue, organoids, or MDSCs using TRIzol Reagent (Molecular Research Center, TR118) according to the manufacturer’s protocol. The High Capacity cDNA Reverse Transcription Kit (Applied Biosystems, 4368813) was used for cDNA synthesis. An amount of 100 ng of RNA per sample was reverse transcribed to yield approximately 2 μg total cDNA and used for the real-time PCR. Pre-designed Real-Time PCR primers were used for the following genes (Thermo Fisher Scientific): mouse-specific CD8 (Mm01182107_g1), granzyme (Mm01313651_m1), Fibronectin (Mm01256744_m1). PCR amplifications were performed in a total volume of 20 μL, containing 20X TaqMan Expression Assay primers, 2X TaqMan Universal Master Mix (Applied Biosystems, TaqMan^®^ Gene Expression Systems) and cDNA template. Each PCR amplification was performed in duplicate wells in a StepOne™ Real-Time PCR System (Applied Biosystems), using the following conditions: 50 °C 2 min, 95 °C 10 min, 95 °C 15 s (denature) and 60 °C 1 min (anneal/extend) for 40 cycles. Fold change was calculated as: (Ct–Ct high) = ntarget, 2ntarget/2nHPRT = fold change where Ct = threshold cycle. The results were expressed as average fold change in gene expression relative to control using GAPDH (human, Hs02786624_g1; mouse, Mm99999915_g1), as an internal control according to Livak and Schmittgen (Methods, 2001, 25(4): 402-8). 

To analyze the human-derived EpCAM-positive (organoid) and -negative (CTL and MDSC) cell fractions by qRT-PCR the following primers were used arginase 1 (Hs00163660_m1), NOS2 (Hs01075529_m1), PD-L1 (Hs00204257_m1), Epcam (Hs00901885_m1), granzyme (Hs00277212_m1), CD8 (Hs00233520_m1), CD15 (Hs01106466_s1), CD14 (Hs02621496_s1), CD66b (Hs00266198_m1), IL4Ra (Hs00965056_m1). For mouse-derived Epcam-positive (organoid) and -negative (CTL and MDSC) fractions, the following primers were used to analyze qRT-PCR: arginase 1 (Mm00475988_m1), NOS2 (Mm00440502_m1), PD-L1 (Mm03048248_m1), Epcam (Mm00493214_m1), granzyme, CD8, CD11b (Mm00434455_m1), GR1 (Mm00439154_m1), Ly6G (Mm04934123_m1), Ly6C (Mm00841873_m1). All primers were purchased from Thermo Fisher Scientific.

### 4.10. NanoString Technologies Digital Spatial Profiling (DSP)

Formalin-fixed paraffin-embedded (FFPE) tissue sections were collected from either mouse experimental groups or human PDAC tissue array (Biochain Institute, Z7020090). FFPE tissue section-mounted slides were incubated with a cocktail of primary antibodies conjugated with DNA oligos via a photocleavable linker together with visible wavelength-imaging reagents. Regions of interest (ROIs) were identified with visible light-based imaging reagents at low-plex to establish overall “architecture” of tumor slices (e.g., image nuclei, immune cell (anti-CD68 for human or anti-CD45 for mouse), and key tumor biomarkers (anti-Pan cytokeratin (PANCK) and anti-smooth muscle actin (SMA)). Selected ROIs were chosen for high-resolution multiplex profiling, and oligos from the selected region are released upon exposure to UV light. Photocleaved oligos were collected via a microcapillary tube and deposited into 96-well microplates, hybridized to 4-color, 6-spot optical barcodes, enabling up to ~1 million digital counts of the protein targets (distributed across all targets) in a single ROI using standard NanoString nCounter instruments. The quantitative data were analyzed and plotted using GraphPad PRISM. 

### 4.11. Luminex Assay

Conditioned media was collected from human and murine tumor-derived organoid cultures and analyzed by Luminex Assay, a high-throughput multiplexing suspension array system (CCHMC, Research Flow Cytometry Core).

### 4.12. Statistical Analysis

Data are expressed as the mean value ± standard error. Analysis was performed using a Student’s *t*-test, one-way or two-way ANOVA, correlation and regression, and differential expression in GraphPad PRISM to determine the differences between groups. Statistical significance was determined when *p* < 0.05.

## 5. Conclusions

Due to late-stage diagnosis and resistance to chemotherapy, pancreatic ductal adenocarcinoma (PDAC) has the lowest five-year survival rate of all cancers in the United States. Programmed death 1 receptor (PD-1)-programmed death ligand 1 (PD-L1) immune checkpoint inhibition has been unsuccessful in clinical trials. Myeloid-derived suppressor cells (MDSCs) are found to be in high abundance in the pancreatic cancer microenvironment. We present a pre-clinical model that may predict the efficacy of combinatorial therapies to improve the outcome of PDAC patients. Here we use mouse- and human-derived autologous pancreatic cancer organoid/immune cell co-cultures and orthotopic transplant mouse model of PDAC to demonstrate that elevated infiltration of polymorphonuclear (PMN)-MDSCs within the PDAC tumor microenvironment inhibit T cell effector function, regardless of PD-1/PD-L1 inhibition. The use of an organoid/immune cell-based platform that can predict patient response to targeted therapies is an unmet therapeutic need. Such an organoid model is required to better understand the development and maintenance of PDAC, and to develop a novel approach for personalized medicine.

## Figures and Tables

**Figure 1 cancers-12-03816-f001:**
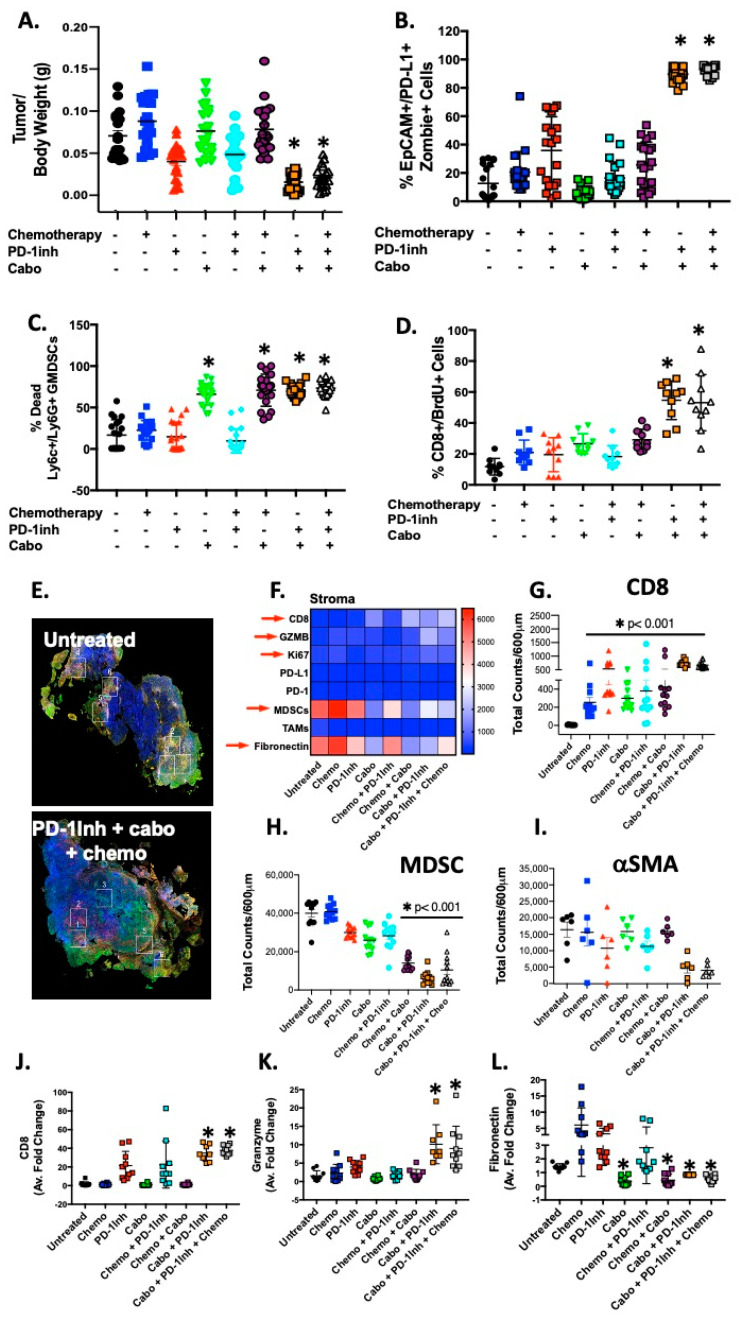
Decreased tumor size in mice treated with combinatorial cabozantinib and anti-PD-1 inhibitor. Changes in (**A**) tumor sizes, (**B**) EpCAM/PD-L1+ve organoid death, (**C**) MDSC death/depletion, and (**D**) CTL proliferation in response to chemotherapy and PD-1Inh with or without cabozantinib treatments. Digital spatial analysis demonstrating (**E**) representative ROIs, and (**F**) immune marker protein expression in mouse experimental groups. Protein expression of (**G**) CD8, (**H**) MDSCs, and (**I**) αSMA quantitative RT-PCR, using RNA isolated from mouse tumors, for the expression of (**J**) CD8, (**K**) granzyme, and (**L**) fibronectin. * *p* < 0.05 compared to untreated; *n* = 10–20 mice per group.

**Figure 2 cancers-12-03816-f002:**
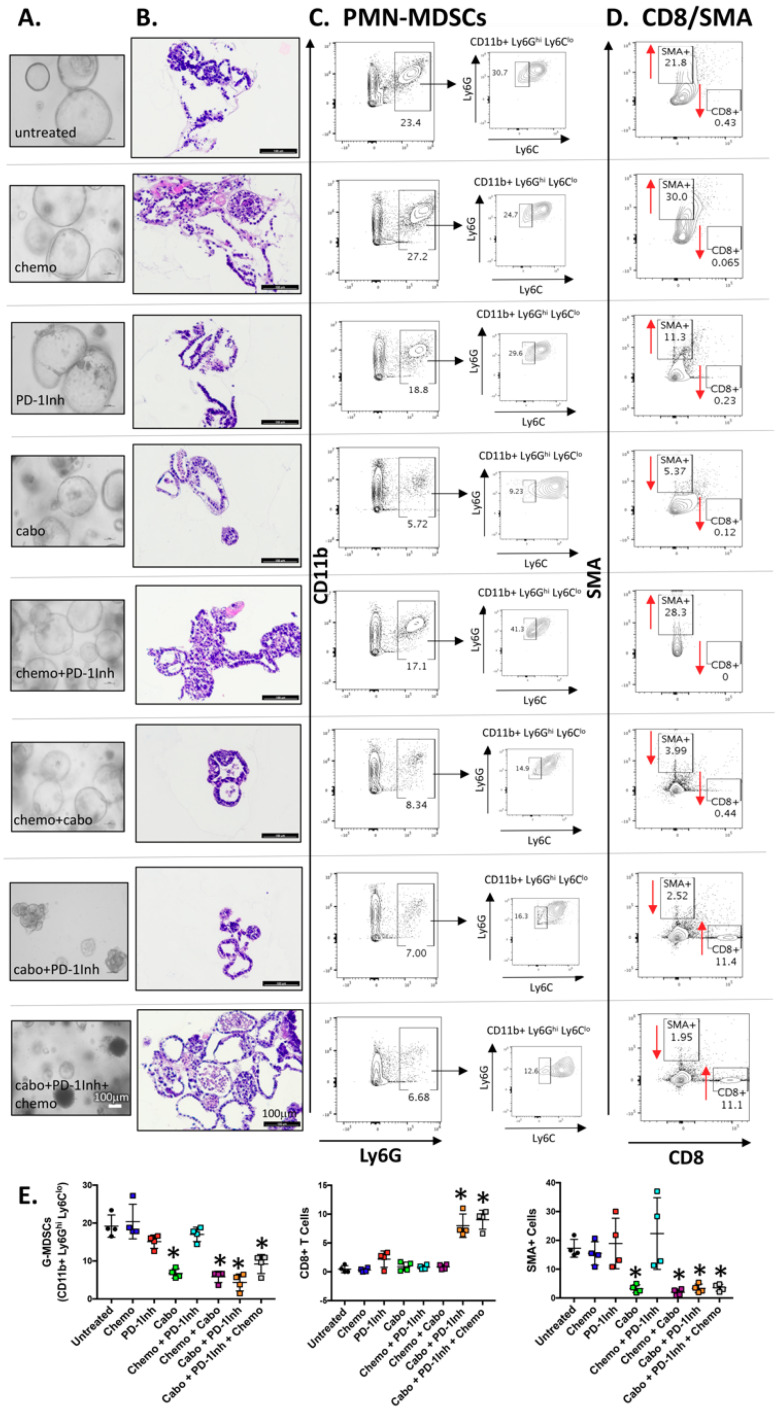
Changes in PMN-MDSC, CD8 and SMA cell compartments in organoids directly derived from mouse tumors in response to experimental treatments. (**A**) Light micrographs of cultured organoids and (**B**) H&E staining of embedded organoids that were derived from mouse tumors in response to experimental treatments. Flow cytometric contour plots demonstrating the changes in (**C**) PMN-MDSC, (**D**) CD8 and SMA cell populations in organoids derived from mouse tumors in response to experimental treatments. Quantification (% cell populations) is shown in (**E**). * *p* < 0.05 compared to untreated; *n* = 10 mice per group.

**Figure 3 cancers-12-03816-f003:**
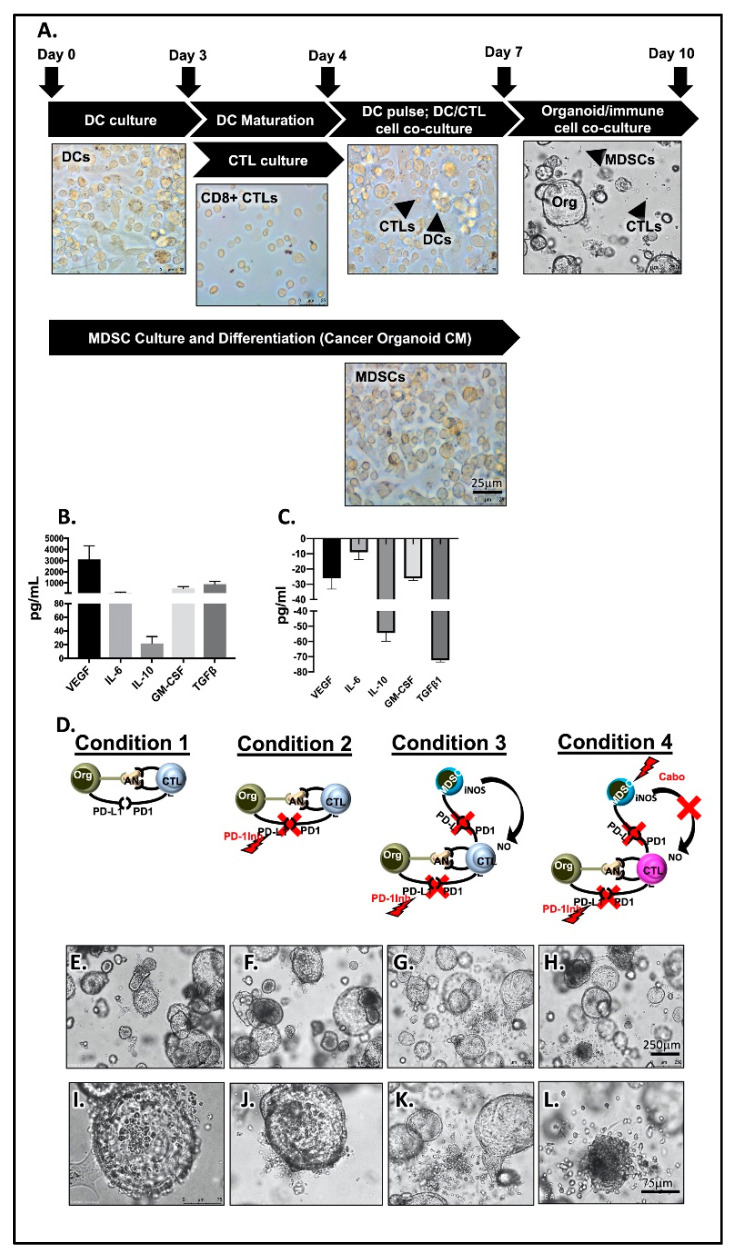
Murine-derived autologous pancreatic cancer organoid/immune cell co-cultures. (**A**) Light micrographs of representative images of immune cells cultured. Luminex assay measuring changes in VEGF, IL-4, IL-10, GM-CSF and TGFβ in conditioned media collected from (**B**) tumor and (**C**) normal organoid cultures. (**D**) Schematic representation of experimental conditions 1, 2, 3 and 4 used in organoid/immune cell co-cultures. Representative light micrographs of organoid/immune cell co-cultures captured in conditions (**E**) 1, (**F**) 2, (**G**) 3, and (**H**) 4. Higher magnifications of conditions (**I**) 1, (**J**) 2, (**K**) 3, and (**L**) 4.

**Figure 4 cancers-12-03816-f004:**
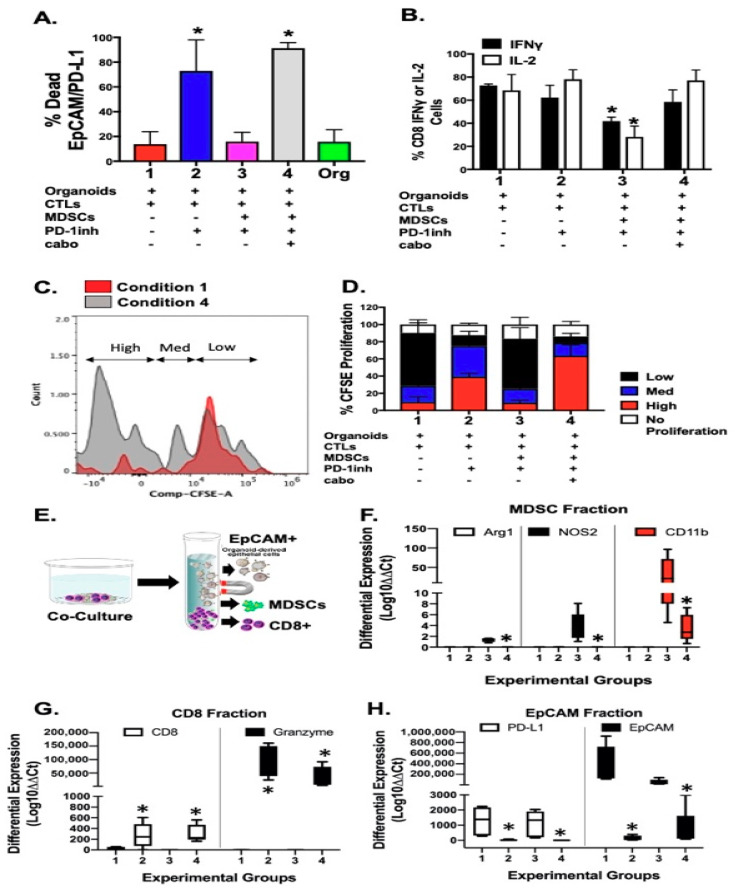
Analysis of murine-derived autologous pancreatic cancer organoid/immune cell co-cultures. Flow cytometric analysis quantifying the percentage of (**A**) zombie-positive (dead) EpCAM+PD-L1+ tumor, and (**B**) CD8+IFNγ+ and CD8+IL-2+ cells in co-cultures. (**C**,**D**) CTL proliferation as measured by CFSE T cell proliferation assay. (**E**) Co-cultures were fractionated into EpCAM+, MDSC+, and CD8+ cell populations by magnetic separation. Quantitative RT-PCR was performed using RNA extracted from (**F**) MDSC, **(G)** CD8, and (**H**) EpCAM fractions. * *p* < 0.05; *n* = 4 individual co-cultures per group.

**Figure 5 cancers-12-03816-f005:**
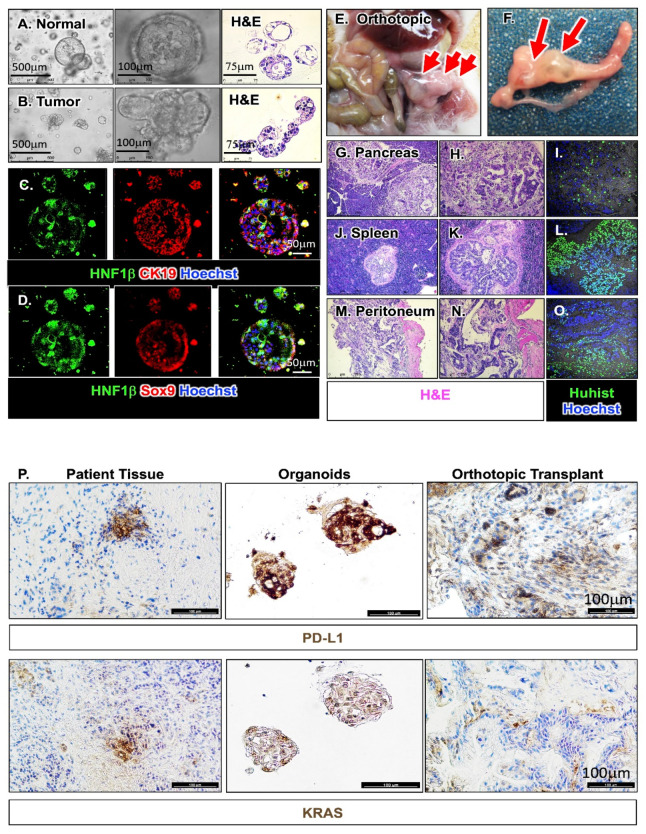
Generation, characterization and orthotopic transplantation of human-derived PDAC organoids. Light micrographs and H&E staining of (**A**) normal and (**B**) PDAC organoids. Immunofluorescence staining of pancreatic organoids for expression of (**C**) HNF1β (green) and cytokeratin 19 (CK19, red) or (**D**) HNF1β (green) and SOX9 (red). Nuclei stain shown by Hoechst in blue. Orthotopic transplantation of metastatic PDAC organoids. (**E**,**F**) Development of pancreatic tumors in NSG mice transplanted with PDAC organoids. H&E stains of (**G**,**H**) pancreas, (**J**,**K**) spleen, and (**M**,**N**) muscle. Immunofluorescence stain of human-specific histone (green) of PDAC lesions within (**I**) pancreas, (**L**) spleen and (**O**) muscle. Nuclei stain shown in Hoechst (blue). Immunohistochemical staining (**P**) of patient tissue, organoids derived from the tissue and orthotopic transplant of the organoids for expression of PD-L1 and KRAS.

**Figure 6 cancers-12-03816-f006:**
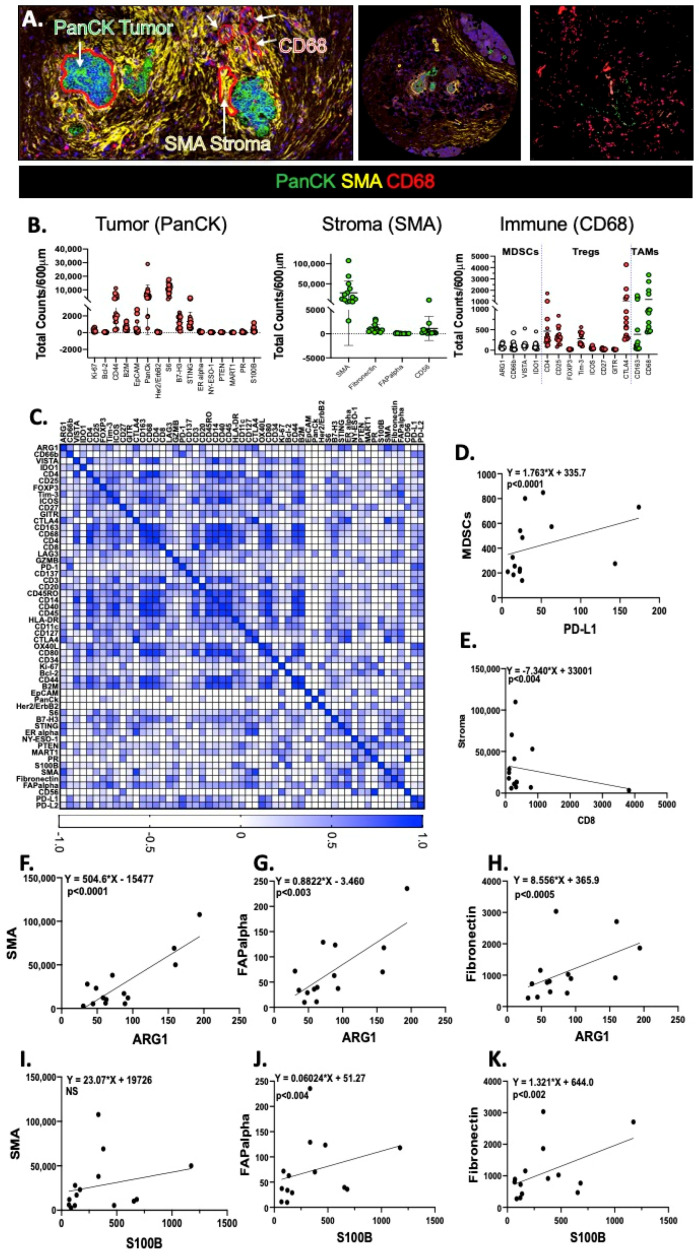
Correlation analysis of patient PDAC tumor tissue array using digital spatial profiling. (**A**) Digital spatial analysis demonstrating representative ROIs, and tumor (PanCK, green), stroma (SMA, yellow) and immune (CD68, red) expression in a patient pancreatic ductal adenocarcinoma tumor tissue array. Scatter plots showing (**B**) tumor, stromal and immune cell components. (**C**) Pearson correlation matrix and correlation between (**D**) MDSCs/PD-L1, (**E**) Stroma/CD8, (**F**) Arg1/SMA, (**G**) Arg1/FAPalpha, (**H**) Arg1/fibronectin, (**I**) S100B/SMA, (**J**) S100B/FAPalpha, and (**K**) S100B/fibronectin. was performed. The intensity of the color at the bottom bar indicates the degree of correlation.

**Figure 7 cancers-12-03816-f007:**
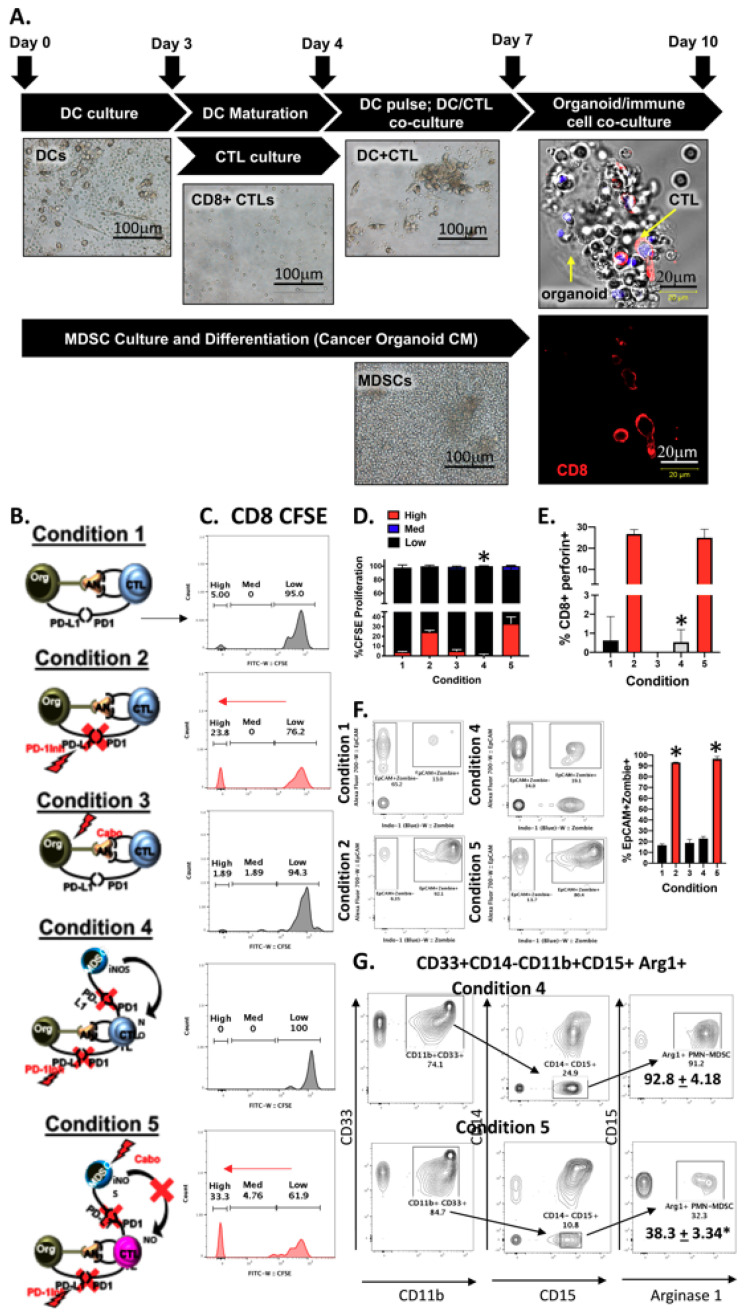
Analysis of patient-derived autologous PDAC organoid/immune cell co-cultures. (**A**) Light micrographs of representative images of patent-derived immune cells cultured with PDAC organoids. (**B**) Schematic representation of experimental conditions 1–5 used in organoid/immune cell co-cultures. (**C**) Histograms generated from CFSE T cell proliferation assays using co-cultures under treatment conditions 1–5. Percent of (**D**) proliferating CTLs, and (**E**) CD8+perforin+-expressing cells in co-cultures under treatment conditions 1–5. (**F**) Representative contour plots were quantified for changes in EpCAM+PD-L1+ zombie (dead) tumor cells within co-cultures under treatment conditions 1–5. (**G**) Representative contour plots were quantified for changes in arginase 1 expression in PMN-MDSCs within co-cultures under treatment conditions 1–5. * *p* < 0.05; *n* = 10 individual patient-derived co-cultures.

**Figure 8 cancers-12-03816-f008:**
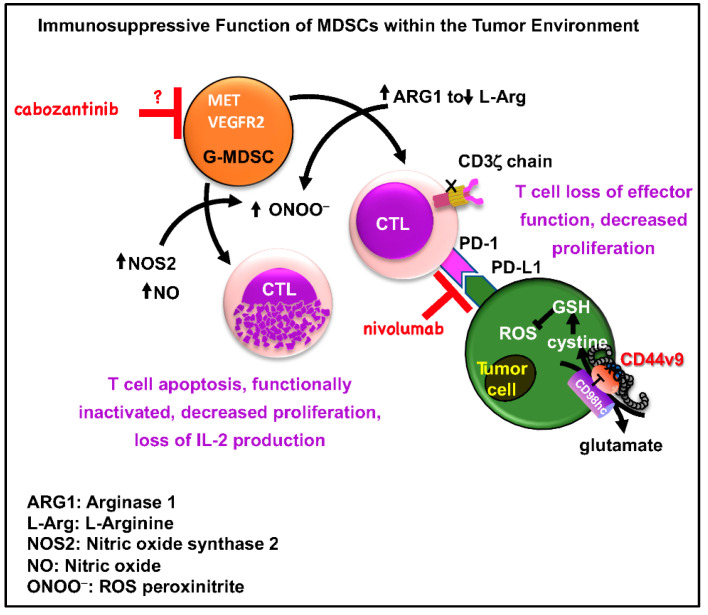
Schematic diagram of MDSC and tumor cell immunosuppression within the pancreatic tumor environment. MDSC secretion of ARG1 decreases L-arginine levels, resulting in a translational blockade of the CD3ζ chain, leading to T cell suppression. In addition, high levels of NOS2 and NO are induced by the activation of MDSCs. Upregulation of both enzymes, increase the production of other reactive oxygen species (ROS) and reactive nitrogen oxide species (RNOS), including O2-, ONOO−, and H_2_O_2_, resulting in T cell apoptosis. ARG1 is upregulated by IL-4, IL-13 and IL-10, whereas NOS2 activity is induced by IFNγ. The inducible cyclooxygenase-2 (COX2) enzyme is critical to the production of prostaglandin E2 (PGE2), and stimulates ARG1 and NOS2 secretion from MDSCs. The tumor cells express CD44v9 that stabilizes xCT to protect cells from ROS-induced apoptosis.

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
