# Peer review of "Murine- and Human-Derived Autologous Organoid/Immune Cell Co-Cultures as Pre-Clinical Models of Pancreatic Ductal Adenocarcinoma"

_cancers, 2020, doi:10.3390/cancers12123816_

Round 1

Reviewer 1 Report

 In the submitted manuscript, Zavros et al., demonstrated that polymorphonuclear myeloid derived suppressor cells, MDSCs are a critical regulator of T cell effector function within the PDAC tumor microenvironment. They elegantly showed that this MDSC-mediated control is independent of PD-1/PD-L1 inhibition. The findings highlighted the potential of the organoid/immune cell co-culture as an effective model to study mouse or human-derived autologous pancreatic cancer. Although the study does not reveal the mechanism of immune- suppressive functionality of MDSCs, yet it sheds light on why immunotherapy is ineffective in significant number of PDAC patients. The models they offer would further help to understand the underlying mechanism in future studies by other groups.

We enjoyed the scientific insight the manuscript offered. Nevertheless, there are few concerns that we believe need to be address.    

Major Comments:

  1. For Figure A, we request the authors to provide ‘tumor mass/body weight ratio’ instead of presenting the data for tumor weight only. This would exclude the body weight bias. In addition, we ask for the measurement of the tumor volume and representative tumor images corresponding to each treatment group. Please show the data points as presented in Figure 1B so that the distribution can be better visualized.
  1. To ensure reproducibility, please mention how many times the in vivo experiment was performed. In general, in vivo studies must be performed twice.
  1. In the result section, the authors mentioned, ‘Decreased tumor weights in the combination treated mice correlated with increased CTL proliferation within the tumor tissue (Figure 1D).’ The conclusion needs to be validated by demonstrating a regression analysis between the two variables (weight vs. cell proliferation). The tumor weight should be replaced by tumor weight/body weight ratio as we indicated earlier. Statistical significance, e.g, R-squared (R2) should be provided as well.

Same requirement is applied for the following statement:

‘In particular, compared to controls, mice treated with either cabo+PD1Inh, or chemo+cabo+PD-1Inh triple treatment, significantly decreased tumor weights correlated with a significant decrease in stromal markers alpha smooth muscle actin, fibronectin and vimentin, and PMN-MDSCs infiltration, with an increase in CD8+ infiltrating cells (Fig. 1E, F).’ Of note, vimentin and SMA are not shown in the heatmap.

  1. In Supplemental Figure 1G, the ROI 11 for stroma shows to be stained for CD45. We speculate that the labelling is inaccurate and it should be for SMA instead.
  1. For Figure 1F, we request to show the quantitative data for immune marker protein expression in addition to the heat map.
  1. The title of Figure 1 does not quite comply with the results demonstrated. The current title is, ‘In vivo sensitization to checkpoint inhibition in mice treated with cabozantinib’. However, checkpoint inhibition was significant when mice were treated by a combination of cabozantinib and PD-1Inh with or without chemotherapy. Same applies for inference from Figure 2.
  1. In Figure 3B, the authors analyzed the secreted cytokines including IL-10, IL-6, GM-CSF, VEGF and TGF in mouse-derived pancreatic cancer organoids. However, to prove their increased secretion, the levels need to be compared to the cytokines secreted from organoids taken from healthy control mice.
  1. For Figure 7E, we request the authors to analyze CD8+ Granzyme B+ cells in addition to quantify CD8+ perforin+ cells by flow cytometry. We ask this since both perforin and granzyme both are equally crucial to mediate the cytotoxic effect of CD8+ cells.

Minor comments:

  1. The font of the labels within the images are not of same size. The images and graphs are also too small to read properly. Please edit following the journal’s guidelines.
  2. In Figure 6A, shouldn’t it be CD68 instead of CD45?
  3. Figure 3C-J needs scale/ magnification added to the micro graphs.

Author Response

Response to Reviewer’s Comments:

We would like to thank the editor and reviewers for the careful and thoughtful review of our manuscript. A detailed response to each comment has been provided below and major changes made to the manuscript have been highlighted in red within the text. 

Reviewer 1:

Major Comments:

  1. Comment:  “For Figure 1A, we request the authors to provide ‘tumor mass/body weight ratio’ instead of presenting the data for tumor weight only. This would exclude the body weight bias. In addition, we ask for the measurement of the tumor volume and representative tumor images corresponding to each treatment group. Please show the data points as presented in Figure 1B so that the distribution can be better visualized.”

Response:  We have modified Figure 1A as suugested by the reviewer. We have calculated the ratio of Tumor weight per gram body weight.

  1. Comment: “To ensure reproducibility, please mention how many times the in vivo experiment was performed. In general, in vivo studies must be performed twice.”

Response:  In order to maintain the reproducibility and sample size, we performed the in vivo experiments two times, and each time, n=10 mice per experimental group.

  1. Comment: “In the result section, the authors mentioned, ‘Decreased tumor weights in the combination treated mice correlated with increased CTL proliferation within the tumor tissue (Figure 1D).’ The conclusion needs to be validated by demonstrating a regression analysis between the two variables (weight vs. cell proliferation). The tumor weight should be replaced by tumor weight/body weight ratio as we indicated earlier. Statistical significance, e.g, R-squared (R2) should be provided as well.”

Response:  We have recalculated the tumor weights to account for body weight in the revised Figure 1A, and have included the statistical significance in Figure 1D.  Supplemental Figure 2 in the revised manuscript also shows the correlation analysis between tumor weight and CTL proliferation.

  1. Comment: “Same requirement is applied for the following statement:

‘In particular, compared to controls, mice treated with either cabo+PD1Inh, or chemo+cabo+PD-1Inh triple treatment, significantly decreased tumor weights correlated with a significant decrease in stromal markers alpha smooth muscle actin, fibronectin and vimentin, and PMN-MDSCs infiltration, with an increase in CD8+ infiltrating cells (Fig. 1E, F).’ Of note, vimentin and SMA are not shown in the heatmap.”

Response: We have included the data of SMA in revised Figure 1I. The vimentin expression mentioned in the results is based on inititial immunofluorescence data. The mouse protein panel from Nanostring DSP did not contain any antibodies for detecting Vimentin.

  1. Comment” “In Supplemental Figure 1G, the ROI 11 for stroma shows to be stained for CD45. We speculate that the labelling is inaccurate and it should be for SMA instead. Modify the figure.”

            Response:  The Figure has been modified.

  1. Comment: “For Figure 1F, we request to show the quantitative data for immune marker protein expression in addition to the heat map.”

Response:  We have plotted the protein expression (Total Counts/600mm) of the immune markers CD8 and MDSCs in the revised Figure 1G and H.

  1. Comment: “The title of Figure 1 does not quite comply with the results demonstrated. The current title is, ‘In vivo sensitization to checkpoint inhibition in mice treated with cabozantinib’. However, checkpoint inhibition was significant when mice were treated by a combination of cabozantinib and PD-1Inh with or without chemotherapy. Same applies for inference from Figure 2.”

Response:  The title of Figures 1 has been changed to “Decreased tumor size in mice treated with combinatorial cabozantinib and anti-PD-1 inhibitor”.

  1. In Figure 3B, the authors analyzed the secreted cytokines including IL-10, IL-6, GM-CSF, VEGF and TGF in mouse-derived pancreatic cancer organoids. However, to prove their increased secretion, the levels need to be compared to the cytokines secreted from organoids taken from healthy control mice.

Response:  We have generated a histogram to address the reviewer’s comment to show the level of secreted cytokines in normal pancreatic organoids. The data is showing negative expression for all secreted cytokines including IL-10, IL-6, GM-CSF, VEGF and TGF.

  1. Comment: “For Figure 7E, we request the authors to analyze CD8+ Granzyme B+ cells in addition to quantify CD8+ perforin+ cells by flow cytometry. We ask this since both perforin and granzyme both are equally crucial to mediate the cytotoxic effect of CD8+ cells.”

Response:  Unfortunatley, the panel of markers that was used for the flow cytometric analysis did not include an antibody to detect changes in granzyme.  We are unable to calculate these changes from our existing data.

Minor comments:

  1. Comment:  “The font of the labels within the images are not of same size. The images and graphs are also too small to read properly. Please edit following the journal’s guidelines.”

Response:  We have tried to increase the font of the labels to make the data clear.

  1. Comment:  In Figure 6A, shouldn’t it be CD68 instead of CD45? CD45 as the major immune marker

Response:  This is corrected in the revised manuscript. 

  1. Comment: “Figure 3C-J needs scale/ magnification added to the micro graphs. ”

Response:  Scale bars are included in all images however, they were not clear in the first submission.  We have increased the font size to make the scale bars clear in the revised manuscript.  Panels Figures 3E-H were captured at 10X and Figures 3I-L were captured at 20X.

Reviewer 2 Report

The authors elaborated a preclinical model with pancreatic cancer organoid and immune cell co-cultures to predict the impact of used inhibitors. The authors showed clearly using mouse- and human-derived autologous pancreatic cancer organoid/immune cell co-cultures  and mouse orthotopic transplant model that depletion of suppressive PMN-MDSCs using cabozantinib resulted in sensitization of PDAC cells to immunotherapy with anti PD-1/PD-L1 antibodies.

In general the manuscript is well elaborated and written, only minor  points require explanation.

  1. Additional use of chemotherapeutic agents together with an MDSC inhibitor and anti-PD-1 antibody only slightly influences the final result. What was the idea of using classic drugs?
  2. The authors used three different TK inhibitors with a similar profile of action (VEGFR, RET, KIT, FLT3) but obtained different results (Fig S4). The authors state (lines 263-267) that sunitinib inhibits STAT3 pathway and regorafenib MAPK pathway. This is not a correct term. Classically, these inhibitors inhibit various TKs. Please discuss the potential target for cabozantinib in MDSCs. It would be interesting to test more specific inhibitors of different TKs or another kinases.
  3. Immunotherapy with anti-PD-1 / PD-L1 antibodies can be effective only in the presence of PD-L1 on the neoplastic cells. How many percent of PDAC patients have this ligand?

I recommend the manuscript to be published after MINOR REVISION.

Author Response

Response to Reviewer’s Comments:

We would like to thank the editor and reviewers for the careful and thoughtful review of our manuscript. A detailed response to each comment has been provided below and major changes made to the manuscript have been highlighted in red within the text.

Reviewer 2:

Major Comments:

  1. Comment:  “Additional use of chemotherapeutic agents together with an MDSC inhibitor and anti-PD-1 antibody only slightly influences the final result. What was the idea of using classic drugs?”

Response:  The additional use of chemotherapeutic agents based on the standard-of-care treatment for patients with PDAC.  Due to the minimal effectiveness of standard-of-care treatments for some patients with PDAC, immunotherapeutics have been used.  In some settings, the addition of immunotherapy combined with chemotherapy significantly improves overall survival in PDAC patients without definitive surgery.  However, in a subset of patients, immunotherapy is ineffective.  Our study suggests, that after standard-of-care chemotherapy, combinatorial therapy using an approach that targets the MDSCs with immunotherapy may be of benefit to PDAC patients that are otherwise resistant to anti-PD-1/PD-L1 inhibition.

  1. Comment:  “The authors used three different TK inhibitors with a similar profile of action (VEGFR, RET, KIT, FLT3) but obtained different results (Fig S4). The authors state (lines 263-267) that sunitinib inhibits STAT3 pathway and regorafenib MAPK pathway. This is not a correct term. Classically, these inhibitors inhibit various TKs. Please discuss the potential target for cabozantinib in MDSCs. It would be interesting to test more specific inhibitors of different TKs or another kinases.”

Response:  Based on the literature we know that cabozantinib, is a tyrosine kinase inhibitor that targets MET, VEGFR2, FLT3, c-KIT and RET. This is discussed in the Discussion section of the manuscript.  We agree that it will be important to identify the specific mechanism by which cabozantinib targets the MDSCs, however, this is beyond the scope of this manuscript and is part of futre studies.

  1. Comment:  “Immunotherapy with anti-PD-1 / PD-L1 antibodies can be effective only in the presence of PD-L1 on the neoplastic cells. How many percent of PDAC patients have this ligand?”

Response:  We have analyzed the TCGA data using CBioPortal analysis website in order to detect the % of PD-L1 expression among TCGA datasets. The result showed that  about 33% patients were positive for PD-L1 expression. This information been added to the Introduction of the revised manuscript.

Reviewer 3 Report

This is a very valuable study, especially the development of an in vitro model to interrogate the role of the immune system in the progression of Pancreatic Ductal Adenocarcinoma. The authors present a compelling mechanism to account for the efficacy of tumor suppression through suppression of MDSCs. This is an invaluable model for preclinical test of the effect of immune-modulatory drugs on patient samples.

Overall the study is well designed and the data are clearly presented. The conclusions of the authors match the data presented.

Minor modifications are required to improve the quality of this elegant study. These are as outlined below:

In most of the figures, scale bars are missing in images. Where scale bar are presented the scale is not legible. The authors should include scale bars and provide within the figure legend the scale. 

Author Response

Response to Reviewer’s Comments:

We would like to thank the editor and reviewers for the careful and thoughtful review of our manuscript. A detailed response to each comment has been provided below and major changes made to the manuscript have been highlighted in red within the text.

Reviewer 3:

Comment:  “Minor modifications are required to improve the quality of this elegant study. These are as outlined below:

In most of the figures, scale bars are missing in images. Where scale bar are presented the scale is not legible. The authors should include scale bars and provide within the figure legend the scale. Add scale bar to Fig 2A, 3 C-J, 5A-D”

Response:  We have increased the font size of the scale bars to make this clear in the revised manuscript.

Reviewer 4 Report

This elegant study presents in vitro organoid model to investigate  the myeloid-derived suppressor cells (MDSCs) as an object  of therapeutic strategy  to treat pancreatic ductal adenocarcinoma (PDAC). Because MDSCs are known to  neutralize CD8 T cells and to deactivate anti-tumor immune response,   authors intended to deplete these cells for better efficacy of programmed death 1 receptor (PD1) – programmed death ligand (PD-L1) immune checkpoint inhibition by nivolumab. For in vivo study  orthoptotic  transplantation of PDAC  cells to mice was used, in vitro model of organoid/immune  cells co-culture based on patient’s and mouse’s tumors.  Depletion of MDSCs by kinase inhibitor cabozantinib  applied together with nivolumab and /or with gemcitabine maximized the efficacy of  nivolumab checkpoint inhibition both in vivo ( murine orthoptotic model ) and in vitro in mouse-derived organoid/immune cell co-culture as well as in patent-derived co-culture.

     The original finding of this study is demonstration of pre-clinical model of pancreatic cancer organoid/immune cell co-culture, which  may be used to predict the patient response to targeted therapies, according to authors conclusion. This model could be also used to study the mechanisms  responsible in  PDAC  development.  

Suggestions and comments:

  1. Experiment protocol could be shown as graphic presentation for better understanding of study .
  2. Presentation of list of abbreviations used in the manuscript could be comfortable for readers and help them to recognize the molecular mechanisms presented in the study. Such list may be added in the beginning or at the end of manuscript.
  3. One of recent clinical studies presents the inhibition of MDSC by combined therapy with PD-1 inhibitor and CXCR4 antagonist in pancreatic cancer, this could be included into Discussion ( Nat.Med.2020).
  4. In Discussion role of IDP was mentioned very shortly as one of mechanisms suppressing antitumor activity. IDO and kynurenic pathway (L-tryptophan, L-kynurenine) are important factors, which block apoptosis in pancreatic cancer cells (Anticancers Agent Med. Chem. 2019).
  5. Discussion is well written and interesting, but could be shortened.

Author Response

Response to Reviewer’s Comments:

We would like to thank the editor and reviewers for the careful and thoughtful review of our manuscript. A detailed response to each comment has been provided below and major changes made to the manuscript have been highlighted in red within the text.

Reviewer 4:

Major Comments:

  1. Comment:  “Experiment protocol could be shown as graphic presentation for better understanding of study . “

Response: We have submitted the graphical abstract as a PDF attachment.

  1. Comment:  “Presentation of list of abbreviations used in the manuscript could be comfortable for readers and help them to recognize the molecular mechanisms presented in the study. Such list may be added in the beginning or at the end of manuscript.”

Response: We have included a list of abbreviations at the end of the manuscript as per Reviewer’s suggestion.

  1. Comment: “One of recent clinical studies presents the inhibition of MDSC by combined therapy with PD-1 inhibitor and CXCR4 antagonist in pancreatic cancer, this could be included into Discussion (Nat.Med.2020).

Response: We have incorporated the reference in the discussion of the revised manuscript on page 26.

  1. In Discussion role of IDP was mentioned very shortly as one of mechanisms suppressing antitumor activity. IDO and kynurenic pathway (L-tryptophan, L-kynurenine) are important factors, which block apoptosis in pancreatic cancer cells (Anticancers Agent Med. Chem. 2019).

Response: This point has been included in the Discussion of the revised manuscript.

  1. Discussion is well written and interesting, but could be shortened.

Response:  We have tried to shorten the Discussion in the revised manuscript.

Round 2

Reviewer 1 Report

We would like to thank the editor and reviewers for the careful and thoughtful review of our manuscript. A detailed response to each comment has been provided below and major changes made to the manuscript have been highlighted in red within the text. 

Reviewer 1:

Major Comments:

  1. Comment:  “For Figure 1A, we request the authors to provide ‘tumor mass/body weight ratio’ instead of presenting the data for tumor weight only. This would exclude the body weight bias. In addition, we ask for the measurement of the tumor volume and representative tumor images corresponding to each treatment group. Please show the data points as presented in Figure 1B so that the distribution can be better visualized.”

Response:  We have modified Figure 1A as suugested by the reviewer. We have calculated the ratio of Tumor weight per gram body weight.

Rev2: We thank the authors for addressing part of our suggestion. However, we would still request to provide the a) measurement of the tumor volume and b) representative tumor images corresponding to each treatment group.

  1. Comment: “To ensure reproducibility, please mention how many times the in vivo experiment was performed. In general, in vivo studies must be performed twice.”

Response:  In order to maintain the reproducibility and sample size, we performed the in vivo experiments two times, and each time, n=10 mice per experimental group.

            Rev2: Accepted. 

  1. Comment: “In the result section, the authors mentioned, ‘Decreased tumor weights in the combination treated mice correlated with increased CTL proliferation within the tumor tissue (Figure 1D).’ The conclusion needs to be validated by demonstrating a regression analysis between the two variables (weight vs. cell proliferation). The tumor weight should be replaced by tumor weight/body weight ratio as we indicated earlier. Statistical significance, e.g, R-squared (R2) should be provided as well.”

Response:  We have recalculated the tumor weights to account for body weight in the revised Figure 1A, and have included the statistical significance in Figure 1D.  Supplemental Figure 2 in the revised manuscript also shows the correlation analysis between tumor weight and CTL proliferation.

Rev2:  We thank the authors for their effort. However, the correlation should be between tumor weight normalized to body weight (tumor/body weight) and CTL and NOT just between tumor weight against CTL. We request to re-do the Supplementary Figure 2 (A-I). 

  1. Comment: “Same requirement is applied for the following statement:

‘In particular, compared to controls, mice treated with either cabo+PD1Inh, or chemo+cabo+PD-1Inh triple treatment, significantly decreased tumor weights correlated with a significant decrease in stromal markers alpha smooth muscle actin, fibronectin and vimentin, and PMN-MDSCs infiltration, with an increase in CD8+ infiltrating cells (Fig. 1E, F).’ Of note, vimentin and SMA are not shown in the heatmap.”

Response: We have included the data of SMA in revised Figure 1I. The vimentin expression mentioned in the results is based on inititial immunofluorescence data. The mouse protein panel from Nanostring DSP did not contain any antibodies for detecting Vimentin.

Rev2: SMA Data inclusion and clarification are cordially accepted. 

  1. Comment: “In Supplemental Figure 1G, the ROI 11 for stroma shows to be stained for CD45. We speculate that the labelling is inaccurate and it should be for SMA instead. Modify the figure.”

             Response:  The Figure has been modified.

Rev2: Thanks. In addition, please check if the ROIs are corrected numbered/labelled. ROI 5 in 1E looks like ROI 3 in 1D; ROI 11 in 1E looks like ROI 6 in 1D.

Author Response

We would like to thank the editor and reviewers for the careful and thoughtful review of our manuscript. A detailed response to each comment has been provided below and changes made to the manuscript have been recorded with track changes within the text.

  1. Reviewer 1

Comment 3: “In the result section, the authors mentioned, ‘Decreased tumor weights in the combination treated mice correlated with increased CTL proliferation within the tumor tissue (Figure 1D).’ The conclusion needs to be validated by demonstrating a regression analysis between the two variables (weight vs. cell proliferation). The tumor weight should be replaced by tumor weight/body weight ratio as we indicated earlier. Statistical significance, e.g, R-squared (R2) should be provided as well.”

First Response:  We have recalculated the tumor weights to account for body weight in the revised Figure 1A, and have included the statistical significance in Figure 1D.  Supplemental Figure 2 in the revised manuscript also shows the correlation analysis between tumor weight and CTL proliferation.

Comment 2:  We thank the authors for their effort. However, the correlation should be between tumor weight normalized to body weight (tumor/body weight) and CTL and NOT just between tumor weight against CTL. We request to re-do the Supplementary Figure 2 (A-I).

Response to Comment 2: Thank you for critically reviewing the manuscript. In order to keep consistency with Fig. 1E we modified the graph (Supplemental Figure 2, page 48) by demonstrating the correlation between tumor weight normalized to body weight (tumor/body weight) and CTL. We also updated the relevant text (page 9, line 147-148; and in the figure legend page 48). 

2.Comment 5: “In Supplemental Figure 1G, the ROI 11 for stroma shows to be stained for CD45. We speculate that the labelling is inaccurate and it should be for SMA instead. Modify the figure.”

             Response:  The Figure has been modified.

Rev2: Thanks. In addition, please check if the ROIs are corrected numbered/labeled. ROI 5 in 1E looks like ROI 3 in 1D; ROI 11 in 1E looks like ROI 6 in 1D.

Response to Rev 2: We have carefully checked the numbering and believe they are correct.